# Chasing Fairness Under Distribution Shift:
# A Model Weight Perturbation Approach

**Zhimeng Jiang**[1]*, **Xiaotian Han**[1]*, **Hongye Jin**[1], **Guanchu Wang**[2], **Rui Chen**[3], **Na Zou**[1], **Xia Hu**[2]
[1]Texas A&M University, [2]Rice University, [3]Samsung Electronics America

## Abstract

Fairness in machine learning has attracted increasing attention in recent years. The fairness methods improving algorithmic fairness for in-distribution data may not perform well under distribution shifts. In this paper, we first theoretically demonstrate the inherent connection between distribution shift, data perturbation, and model weight perturbation. Subsequently, we analyze the sufficient conditions to guarantee fairness (i.e., low demographic parity) for the target dataset, including fairness for the source dataset, and low prediction difference between the source and target datasets for each sensitive attribute group. Motivated by these sufficient conditions, we propose robust fairness regularization (RFR) by considering the worst case within the model weight perturbation ball for each sensitive attribute group. We evaluate the effectiveness of our proposed RFR algorithm on synthetic and real distribution shifts across various datasets. Experimental results demonstrate that RFR achieves better fairness-accuracy trade-off performance compared with several baselines. The source code is available at https://github.com/zhimengj0326/RFR_NeurIPS23.

## 1 Introduction

Previous research [1–4] has shown that a classifier trained on a specific source distribution will perform worse when testing on a different target distribution, due to distribution shift. Recently, many studies have focused on investigating the impact of distribution shift on machine learning models, where fairness performance degradation is even more significant than that of prediction performance [5]. The sensitivity of fairness over distribution shift challenges machine learning models in high stake applications, such as criminal justice [6], healthcare [7], and job marketing [8]. Thus the transferability of the fairness performance under distribution shift is a crucial consideration for real-world applications.

To achieve the fairness of the model (already achieved fairness on the source dataset ) on the target dataset, we first reveal that distribution shift is equivalent to model weight perturbation, and then seek to achieve fairness under distribution shift via model weight perturbation. Specifically, *i)* we reveal the inherent connection between distribution shift, data perturbation, and model weight perturbation. We theoretically demonstrate that any distribution shift can be equivalent to data perturbation and model weight perturbation in terms of loss value. In other words, the effect of distribution shift on model training can be attributed to data or model perturbation. *ii)* Given the established connection between the distribution shift and model weight perturbation, we next tackle fairness under the distribution shift problem. We first investigate demographic parity relation between source and target datasets. Achieving fair prediction (e.g., low demographic parity) in the source dataset is insufficient for fair prediction in the target dataset. More importantly, the average prediction difference between source and target datasets with the same sensitive attribute also matters for achieving fairness in target datasets.

---

*Equal contribution.

37th Conference on Neural Information Processing Systems (NeurIPS 2023).

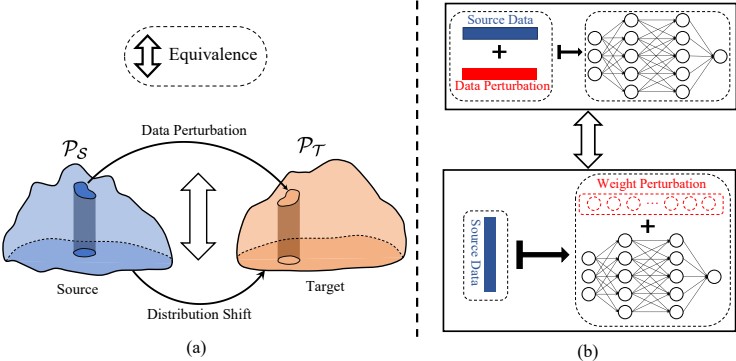

Figure 1: The overview of distribution shift understanding. The left part demonstrates distribution shift can be transformed as data perturbation, while the right part shows that data perturbation and model weight perturbation are equivalent.

Motivated by the established equivalence connection, we propose robust fairness regularization (RFR) to enforce average prediction robustness over model weight. In this way, the well-trained model can tackle the distribution shift problem in terms of demographic parity. Considering the expensive computation for the inner maximization problem in RFR, we accelerate RFR by obtaining the approximate closed-form model weight perturbation using first-order Taylor expansion. In turn, an efficient RFR algorithm, trading the inner maximization problem into two forward and backward propagations for each model update, is proposed to achieve robust fairness under distribution shift. Our contributions are highlighted as follows:

- We theoretically reveal the inherent connection between distribution shift, data perturbation, and model weight perturbation. In other words, distribution shift and model perturbation are equivalent in terms of loss value for any model architecture and loss function.

- We first analyze the sufficient conditions to guarantee fairness transferability under distribution shift. Based on the established connection, we propose RFR explicitly pursuing sufficient conditions for robust fairness by considering the worst case of model perturbation.

- We evaluate the effectiveness of RFR on various real-world datasets with synthetic and real distribution shifts. Experiments results demonstrate that RFR can mostly achieve better fairness-accuracy tradeoff with both synthetic and real distribution shifts.

## 2 Understanding Distribution Shift

In this section, we first provide the notations used in this paper. And then, we theoretically understand the relations between distribution shift, data perturbation, and model weight perturbation, as shown in Figure 1.

### 2.1 Notations

We consider source dataset $\mathcal{D}_\mathcal{S}$ and target dataset $\mathcal{D}_\mathcal{T}$, defined as a probability distribution $\mathcal{P}_\mathcal{S}$ and $\mathcal{P}_\mathcal{T}$ for samples $S \in \mathcal{S}$ and $T \in \mathcal{T}$, respectively. Each sample defines values for three random variables: features $X$ with arbitrary domain $\mathcal{X}$, binary sensitive attribute $A \in \mathcal{A} = \{0, 1\}$, and label $Y$ arbitrary domain $\mathcal{Y}$, i.e., $\mathcal{S} = \mathcal{T} = \mathcal{X} \times \mathcal{A} \times \mathcal{Y}$. We denote $\delta$ as data perturbation on source domain $\mathcal{S}$, where $\delta_X(X)$ and $\delta_Y(Y)$ data perturbation of features and labels. Using $\mathcal{P}(\cdot)$ to denote the space of probability distributions over some domain, we denote the space of distributions over examples as $\mathcal{P}(\mathcal{S})$. We use $||\cdot||_p$ as $L_p$ norm. Let $f_\theta(x)$ be the output of the neural networks parameterized with $\theta$ to approximate the true label $y$. We define $l(f_\theta(x), y)$ as the loss function for neural network training, and the optimal model parameters $\theta^*$ training on source data $\mathcal{S}$ is given by $\theta^* = \arg\min_\theta \mathcal{R}_\mathcal{S}$, where

$\mathcal{R}_\mathcal{S} = \mathbb{E}_{(X,Y) \sim \mathcal{P}_\mathcal{S}}[l(f_\theta(X), Y)]$. Since the target distribution $\mathcal{P}_\mathcal{T}$ maybe be different with source distribution $\mathcal{P}_\mathcal{S}$, the well-trained model $f_{\theta^*}(\cdot)$ trained on soure dataset $\mathcal{S}$ typically does not perform well in target dataset $\mathcal{T}$.

## 2.2 Distribution Shift is Data Perturbation

Deep neural networks are immensely challenged by data quality or dynamic environments [9, 10]. The well-trained deep neural network model may not perform well if existing feature/label noise in training data or distribution shifts among training and target environments. In this subsection, we reveal the inherent equivalence between distribution shift and data perturbation for any neural networks $f_\theta(\cdot)$, source distribution $\mathcal{P}_\mathcal{S}$, and target distribution $\mathcal{P}_\mathcal{T}$ using optimal transport. We first provide the formal definition of optimal transport as follows:

**Definition 1** (Optimal Transport [11]). *Considering two distributions with probability distribution $P_\mathcal{S}$ and $P_\mathcal{T}$, and cost function moving from $s$ to $t$ as $c(s,t)$, optimal transport between probability distribution $P_\mathcal{S}$ and $P_\mathcal{T}$ is given by*

$$\gamma^*(s,t) = \arg \inf_{\gamma \in \Gamma(P_\mathcal{S}, P_\mathcal{T})} \iint c(s,t)\gamma(s,t)\mathrm{d}s\mathrm{d}t, \tag{1}$$

*where the distribution set $\Gamma(P_\mathcal{S}, P_\mathcal{T})$ is the collection of all possible transportation plans and given by $\Gamma(P_\mathcal{S}, P_\mathcal{T}) = \left\{ \gamma(s,t) > 0, \int \gamma(s,t)\mathrm{d}t = P_\mathcal{S}(s), \int \gamma(s,t)\mathrm{d}s = P_\mathcal{T}(t) \right\}$. In other words, the distribution set consists of all possible joint distributions with margin distribution $P_\mathcal{S}$ and $P_\mathcal{T}$.*

Based on the definition of optimal transport, we demonstrate that distribution shift is equivalent to data perturbation in the following theorem:

**Theorem 2.1.** *For any two different distributions with probability distribution $P_\mathcal{S}$ and $P_\mathcal{T}$, adding data perturbation $\delta$ [2] on source data $S$ can make perturbated source data and target data with the same distribution, where the distribution of data perturbation $\delta$ is given by*

$$\mathcal{P}(\delta) = \int_\mathcal{S} \gamma^*(s, s+\delta)\mathrm{d}s. \tag{2}$$

*Additionally, for any positive $p > 0$, data perturbation $\delta$ with minimal power $\mathbb{E}[||\delta||_p^p]$ is given by Eq.(2) if optimal transport plan $\gamma^*(\cdot, \cdot)$ is calculated based on Eq. (1) with cost function $c(s,t) = ||s-t||_p^p$.*

**Proof sketch.** Given two different distributions, there are many possible perturbations to move one distribution to another. Optimal transport can select the optimal feasible perturbations or transportation plan in terms of specific objectives (e.g., perturbations power). Given the optimal transportation plan, we can derive the distribution of perturbations based on basic probability theory.

Theorem 2.1 demonstrates the equivalence between distribution shift and data perturbation, where data perturbation distribution is dependent on optimal transport between source and target distribution. The intuition is that such distribution shift can be achieved via optimal transportation (i.e., data perturbation). Based on Theorem 2.1, we have the following Corollary 2.2 on the equivalent of neural network behavior for source and target data:

**Corollary 2.2.** *Given source and target datasets with probability distribution $\mathcal{P}_\mathcal{S}$ and $\mathcal{P}_\mathcal{T}$, there exists data perturbation $\delta$ so that the training loss of any neural network $f_\theta(\cdot)$ for target distribution equals that for source distribution with data perturbation $\delta$, i.e.,*

$$\mathbb{E}_{(X,Y)\sim\mathcal{P}_\mathcal{T}}[l(f_\theta(X), Y)] = \mathbb{E}_{\delta_X(X), \delta_Y(Y)}\mathbb{E}_{(X,Y)\sim\mathcal{P}_\mathcal{S}}[l(f_\theta(X + \delta_X(X)), Y + \delta_Y(Y))]. \tag{3}$$

In other words, for the model trained with loss minimization on source data, the deteriorated performance on the target dataset stems from the perturbation of features and labels. The proof sketch is based on Theorem 2.1 since adding a perturbation on the source dataset can be consistent with the target dataset distribution. Therefore, the conclusion can hold for any loss function that is only dependent on the dataset.

## 2.3 Data Perturbation Equals Model Weight Perturbation

Although we understand that the distribution shift can be attributed to the data perturbation of features and labels in source data, it is still unclear how to tackle the distribution shift issue. A natural solution

---

[2] Data perturbation $\delta$ includes the perturbation for features $\delta_X(X)$ and labels $\delta_Y(Y)$.

is to adopt adversarial training to force the well-trained model to be robust over data perturbation. However, it is complicated to generate data perturbation on features and labels simultaneously, and many well-developed adversarial training methods are mainly designed for adversarial feature perturbation. Fortunately, we show that model weight perturbation is equivalent to data perturbation by Theorem 2.3.

**Theorem 2.3.** *Considering the source dataset with distribution $\mathcal{P}_\mathcal{S}$, suppose the source dataset is perturbed with data perturbation $\delta$, and the neural network is given by $f_\theta(\cdot)$, for general case, there exists model weight perturbation $\Delta\theta$ so that the training loss on perturbed source dataset is the same with that for model weight perturbation $\Delta\theta$ on source distribution:*

$$\mathbb{E}_{\delta_X(X), \delta_Y(Y)} \mathbb{E}_{(X,Y) \sim \mathcal{P}_\mathcal{S}}[l(f_\theta(X + \delta_X(X)), Y + \delta_Y(Y))] = \mathbb{E}_{(X,Y) \sim \mathcal{P}_\mathcal{S}}[l(f_{\theta+\Delta\theta}(X), Y)]. \quad (4)$$

**Proof sketch.** The loss under data perturbation and weight perturbation can be analyzed using first-order Tayler expansion. The critical step is to find the condition of the equivalence for the first order term of data perturbation and weight perturbation. Fortunately, the existence of such conditions can be easily proved using linear algebra.

Theorem 2.3 demonstrates that the training loss on perturbed data distribution (but fixed model weight) equals the training loss on perturbed model weight (but original data distribution). In other words, the training loss fluctuation from data perturbation can equal model weight perturbation. Furthermore, we conclude that chasing a robust model over data perturbation can be achieved via model weight perturbation, i.e., finding a "flattened" local minimum in terms of the target objective is sufficient for robustness.

## 3 Methodology

In this section, we first analyze the sufficient condition to achieve robust fairness over distribution shift in terms of demographic parity. Based on the analysis, we propose a simple yet effective robust fairness regularization via explicit adopting group model weight perturbation. Note that the proposed robust fairness regularization involves a computation-expensive maximization problem, we further accelerate model training via first-order Taylor expansion.

### 3.1 Robust Fairness Analysis

We consider binary sensitive attribute $A \in \{0, 1\}$ and demographic parity as fairness metric, i.e., the average prediction gap of model $f_\theta(\cdot)$ for different sensitive attribute groups in target dataset $\Delta DP_\mathcal{T} = |\mathbb{E}_{\mathcal{T}_0}[f_\theta(\mathbf{x})] - \mathbb{E}_{\mathcal{T}_1}[f_\theta(\mathbf{x})]|$, where $\mathcal{T}_0$ and $\mathcal{T}_1$ represent the target datasets for sensitive attribute $A = 0$ and $A = 1$, respectively. However, only source dataset $\mathcal{S}$ is available for neural network training. In other words, even though demographic parity on source dataset $\Delta DP_\mathcal{S} = |\mathbb{E}_{\mathcal{S}_0}[f_\theta(\mathbf{x})] - \mathbb{E}_{\mathcal{S}_1}[f_\theta(\mathbf{x})]|$ is low, demographic parity on target dataset $\Delta DP_\mathcal{T}$ may not be guaranteed to be low due to distribution shift.

To investigate robust fairness over distribution shift, we try to reveal the connection between demographic parity for source and target datasets. We bound the demographic parity difference for source and target datasets as follows:

$$
\begin{aligned}
DP_\mathcal{T} &\overset{(a)}{\leq} DP_\mathcal{S} + \left| |\mathbb{E}_{\mathcal{T}_0}[f_\theta(\mathbf{x})] - \mathbb{E}_{\mathcal{T}_1}[f_\theta(\mathbf{x})]| - |\mathbb{E}_{\mathcal{S}_0}[f_\theta(\mathbf{x})] - \mathbb{E}_{\mathcal{S}_1}[f_\theta(\mathbf{x})]| \right| \\
&\overset{(b)}{\leq} DP_\mathcal{S} + |\mathbb{E}_{\mathcal{S}_0}[f_\theta(\mathbf{x})] - \mathbb{E}_{\mathcal{T}_0}[f_\theta(\mathbf{x})]| + |\mathbb{E}_{\mathcal{S}_1}[f_\theta(\mathbf{x})] - \mathbb{E}_{\mathcal{T}_1}[f_\theta(\mathbf{x})]|, \quad (5)
\end{aligned}
$$

where inequality (a) and (b) hold due to $a - b \leq |a - b|$ and $||a - b| - |a' - b'|| \leq |a - a'| + |b - b'|$, respectively, for any $a, a', b, b'$. In other words, in order to minimize demographic parity for target dataset $DP_\mathcal{T}$, the objective of $DP_\mathcal{S}$ minimization is insufficient. The minimization of prediction difference for source and target datasets given sensitive attribute groups $A = 0$ and $A = 1$, defined as $\Delta_0 = |\mathbb{E}_{\mathcal{S}_0}[f_\theta(\mathbf{x})] - \mathbb{E}_{\mathcal{T}_0}[f_\theta(\mathbf{x})]|$ and $\Delta_1 = |\mathbb{E}_{\mathcal{S}_1}[f_\theta(\mathbf{x})] - \mathbb{E}_{\mathcal{T}_1}[f_\theta(\mathbf{x})]|$, are also beneficial to achieve robust fairness over distribution shift in terms of demographic parity.

The bound in Eq. (5) is tight when both condition (a) $DP_\mathcal{S} \leq DP_\mathcal{T}$ and (b) maximum and minimum of value set $\min\{\mathbb{E}_{\mathcal{S}_0}[f_\theta(\mathbf{x})], \mathbb{E}_{\mathcal{T}_0}[f_\theta(\mathbf{x})]\}, \min\{\mathbb{E}_{\mathcal{S}_1}[f_\theta(\mathbf{x})], \mathbb{E}_{\mathcal{T}_1}[f_\theta(\mathbf{x})]\}$ both are from source or target distribution. Even though conditions (a) and (b) may not hold for the neural network model, we

would like to that our goal is not to obtain a tight bound for demographic parity on target distribution. Instead, we aim to find sufficient conditions to guarantee low demographic parity and such low demographic parity can be achieved in model training without any target distribution information. The proposed upper bound Eq. (5) actually reveals sufficient conditions, which can be achieved by our proposed RFR algorithm.

## 3.2 Robust Fairness Regularization

Motivated by Section 3.1 and distribution shift understanding in Section 2, we develop a robust fairness regularization to achieve robust fairness over distribution shift. Section 3.1 demonstrates that fair prediction on the target dataset requires fair prediction on the source dataset and low prediction difference between the source and target dataset for each sensitive attribute, i.e., low $\Delta_0 = |\mathbb{E}_{\mathcal{S}_0}[f_\theta(\mathbf{x})] - \mathbb{E}_{\mathcal{T}_0}[f_\theta(\mathbf{x})]|$ and $\Delta_1 = |\mathbb{E}_{\mathcal{S}_1}[f_\theta(\mathbf{x})] - \mathbb{E}_{\mathcal{T}_1}[f_\theta(\mathbf{x})]|$. Based on Theorem 2.3, there exists $\epsilon_0$ so that the following equality holds:

$$\Delta_0 = |\mathbb{E}_{\mathcal{S}_0}[f_\theta(\mathbf{x})] - \mathbb{E}_{\epsilon_0}\mathbb{E}_{\mathcal{S}_0}[f_{\theta+\epsilon_0}(\mathbf{x})]|. \tag{6}$$

Note that the distribution shift is unknown, we consider the worst case for model weight perturbation $\epsilon_0$ within $L_p$-norm perturbation ball with radius $\rho$ as follows:

$$\Delta_0 = |\mathbb{E}_{\mathcal{S}_0}[f_\theta(\mathbf{x})] - \mathbb{E}_{\epsilon_0}\mathbb{E}_{\mathcal{S}_0}[f_{\theta+\epsilon_0}(\mathbf{x})]| \leq \max_{\|\epsilon_0\|_p \leq \rho} |\mathbb{E}_{\mathcal{S}_0}[f_{\theta+\epsilon_0}(\mathbf{x})] - \mathbb{E}_{\mathcal{S}_0}[f_\theta(\mathbf{x})]|, \tag{7}$$

where $\|\cdot\|_p$ represents $L_p$ norm, $\rho$ and $p$ are hyperparameters. Note that the feasible region of model weight perturbation $\epsilon_0$ is symmetric, and the neural network prediction is locally linear around parameter $\theta$, we have $\mathbb{E}_{\mathcal{S}_0}[f_{\theta+\epsilon_0}(\mathbf{x})] - \mathbb{E}_{\mathcal{S}_0}[f_\theta(\mathbf{x})] \approx -\left(\mathbb{E}_{\mathcal{S}_0}[f_{\theta-\epsilon_0}(\mathbf{x})] - \mathbb{E}_{\mathcal{S}_0}[f_\theta(\mathbf{x})]\right)$ due to the local linearity. In other words, the absolute operation can be removed if we consider the maximation problem in a symmetric feasible region since there are always non-negative value for the pair perturbation $\epsilon_0$ and $-\epsilon_0$. Therefore, we can further bound $\Delta_0$ as

$$\begin{aligned}
\Delta_0 &\leq \max_{\|\epsilon_0\|_p \leq \rho} |\mathbb{E}_{\mathcal{S}_0}[f_{\theta+\epsilon_0}(\mathbf{x})] - \mathbb{E}_{\mathcal{S}_0}[f_\theta(\mathbf{x})]| \\
&\approx \max_{\|\epsilon_0\|_p \leq \rho} \mathbb{E}_{\mathcal{S}_0}[f_{\theta+\epsilon_0}(\mathbf{x})] - \mathbb{E}_{\mathcal{S}_0}[f_\theta(\mathbf{x})] \stackrel{\triangle}{=} \mathcal{L}_{RFR,\mathcal{S}_0},
\end{aligned} \tag{8}$$

Similarly, we can bound the prediction difference for source and target distribution with sensitive group $A = 1$ as follows:

$$\Delta_1 \leq \max_{\|\epsilon_1\|_p \leq \rho} \mathbb{E}_{\mathcal{S}_1}[f_{\theta+\epsilon_1}(\mathbf{x})] - \mathbb{E}_{\mathcal{S}_1}[f_\theta(\mathbf{x})] \stackrel{\triangle}{=} \mathcal{L}_{RFR,\mathcal{S}_1}. \tag{9}$$

Therefore, demographic parity for source and target distribution relation is given by $DP_\mathcal{T} \leq DP_\mathcal{S} + \mathcal{L}_{RFR,\mathcal{S}_0} + \mathcal{L}_{RFR,\mathcal{S}_1}$, we propose *robust fairness regualarization* (RFR) to achieve robust fairness as

$$\mathcal{L}_{RFR} = \mathcal{L}_{RFR,\mathcal{S}_0} + \mathcal{L}_{RFR,\mathcal{S}_1}. \tag{10}$$

It is worth noting that our proposed RFR is agnostic to the training loss function and model architectures. Additionally, we follow [12] to accelerate model training using sharpness-aware minimization via trading maximization problem can be simplified as two forward and two backward propagations. More details on training acceleration are in Appendix E.

## 3.3 The Proposed Method

In this subsection, we introduce how to use the proposed RFR to achieve robust fairness, i.e., the fair model (low $\Delta DP_\mathcal{S}$) trained on the source dataset will also be fair on the target dataset (low $\Delta DP_\mathcal{T}$). Considering binary sensitive attribute $A \in \{0, 1\}$ and binary classification problem $Y \in \{0, 1\}$, the classification loss is denoted as

$$\mathcal{L}_{CLF} = \mathbb{E}_{\mathcal{S}}[-Y f_\theta(X) - (1-Y)(1 - f_\theta(X))]. \tag{11}$$

To achieve fairness, we consider demographic parity as fairness regularization, i.e.,

$$\mathcal{L}_{DP} = |\mathbb{E}_{\mathcal{S}_0}[f_\theta(\mathbf{x})] - \mathbb{E}_{\mathcal{S}_1}[f_\theta(\mathbf{x})]|, \tag{12}$$

Table 1: Performance Comparison with Baselines on Synthetic Dataset. $(\alpha, \beta)$ control distribution shift intensity, and $(0, 1)$ represents no distribution shift. The best/second-best results are highlighted in **boldface**/underlined, respectively.

| $(\alpha, \beta)$ | Methods | Adult | | | ACS-I | | | ACS-E | | |
|---|---|---|---|---|---|---|---|---|---|---|
| | | Acc (%) ↑ | $\Delta_{DP}$ (%) ↓ | $\Delta_{EO}$ (%) ↓ | Acc (%) ↑ | $\Delta_{DP}$ (%) ↓ | $\Delta_{EO}$ (%) ↓ | Acc (%) ↑ | $\Delta_{DP}$ (%) ↓ | $\Delta_{EO}$ (%) ↓ |
| (1.0, 2.0) | MLP | 82.09±0.05 | 15.11±0.04 | 14.33±0.05 | 77.95±0.52 | 3.51±0.59 | 3.77±0.55 | 80.95±0.10 | 1.10±0.06 | 1.43±0.06 |
| | REG | 80.60±0.05 | 3.79±0.06 | 3.27±0.08 | 77.77±0.09 | 2.28±0.32 | 2.59±0.23 | 80.44±0.07 | 0.86±0.09 | 1.05±0.10 |
| | ADV | 78.80±0.68 | 0.83±0.26 | 0.79±0.14 | 75.72±0.63 | 1.96±0.38 | 2.00±0.35 | 79.39±0.15 | 1.09±0.26 | 0.95±0.26 |
| | FCR | 79.06±0.09 | 9.98±0.06 | 9.47±0.07 | 76.99±0.47 | 2.94±0.34 | 2.95±0.28 | 79.74±0.11 | 0.97±0.21 | 1.00±0.22 |
| | RFR | 78.84±0.09 | **0.44±0.05** | **0.12±0.06** | 74.15±0.81 | **1.84±0.27** | **1.60±0.33** | 80.08±0.08 | **0.71±0.10** | **0.06±0.11** |
| (1.5, 3.0) | MLP | 82.05±0.05 | 15.16±0.09 | 14.33±0.09 | 77.85±0.25 | 3.73±0.53 | 3.70±0.56 | 80.42±0.10 | 1.14±0.07 | 1.10±0.07 |
| | REG | 80.64±0.08 | 3.74±0.11 | 3.23±0.10 | 77.87±0.18 | 2.25±0.28 | 2.37±0.27 | 80.21±0.13 | **0.72±0.04** | 0.75±0.03 |
| | ADV | 78.71±0.41 | 1.07±0.87 | 0.87±0.96 | 75.79±0.68 | 2.22±0.53 | 2.44±0.48 | 79.58±0.13 | 1.07±0.19 | 1.26±0.18 |
| | FCR | 79.05±0.12 | 10.01±0.07 | 9.51±0.06 | 77.06±0.68 | 3.39±0.33 | 3.10±0.36 | 79.59±0.26 | 1.17±0.24 | 1.08±0.23 |
| | RFR | 78.91±0.03 | **0.46±0.10** | **0.16±0.09** | 74.19±0.58 | **1.82±0.29** | 2.17±0.32 | 80.47±0.03 | **0.72±0.04** | **0.71±0.05** |
| (3.0, 6.0) | MLP | 82.07±0.05 | 15.23±0.14 | 14.45±0.15 | 77.89±0.45 | 3.35±0.36 | 3.47±0.41 | 80.30±0.04 | 1.17±0.04 | 1.13±0.04 |
| | REG | 80.62±0.07 | 3.72±0.05 | 3.21±0.04 | 78.19±0.12 | **1.60±0.48** | **1.84±0.44** | 80.36±0.09 | **0.70±0.09** | 0.68±0.11 |
| | ADV | 78.97±0.49 | **1.28±0.74** | **1.09±0.50** | 75.71±0.68 | 2.28±0.39 | 2.24±0.41 | 79.66±0.16 | 1.34±0.14 | 1.16±0.13 |
| | FCR | 79.03±0.13 | 10.00±0.05 | 9.50±0.05 | 76.71±0.39 | 2.97±0.34 | 3.28±0.31 | 79.89±0.22 | 1.06±0.14 | 1.14±0.18 |
| | RFR | 80.15±0.07 | 1.75±0.15 | 1.30±0.14 | 74.22±0.56 | 1.80±0.26 | 1.89±0.24 | 80.28±0.12 | 0.74±0.04 | **0.51±0.04** |

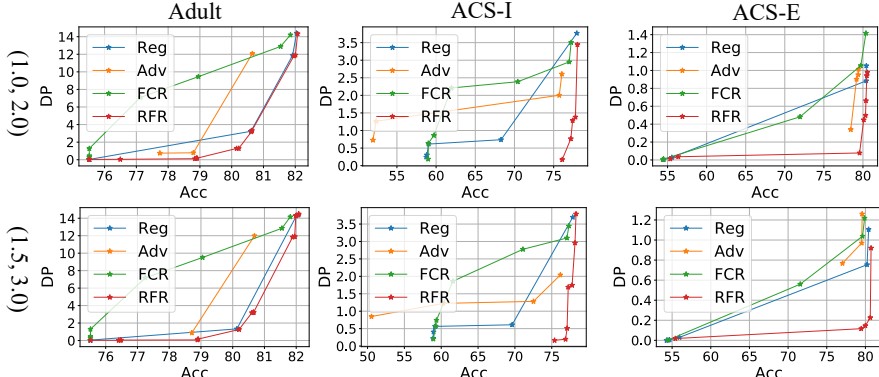

Figure 2: The fairness (DP) and prediction (Acc) trade-off performance on three datasets with different synthetic distribution shifts. The units for x- and y-axis are percentages (%).

Additionally, we also adopt $\mathcal{L}_{RFR}$ as robust fairness regularization. The overall objective is given as follows:

$$\mathcal{L}_{all} = \mathcal{L}_{CLF} + \lambda \cdot \left( \mathcal{L}_{DP} + \mathcal{L}_{RFR} \right), \tag{13}$$

where $\lambda$ is a hyperparameter to balance the model performance and fairness. $\mathcal{L}_{CLF}$ is the loss function for the downstream task, $\mathcal{L}_{DP}$ is to ensure the demographic parity on the source dataset, and $\mathcal{L}_{RFR}$ is to ensure the transferability of fairness from the source dataset to the target dataset.

In the proposed RFR algorithm, we directly apply approximation to accelerate model training. Note that the optimal model weight perturbation is dependent on the current model weight $\theta$, two forward and backward propagations are required to calculate the final gradient.

## 4 Experiments

In this section, we conduct experiments to evaluate the effectiveness of our proposed RFR, aiming to answer the three research questions. **Q1**: How effective is RFR to achieve fairness under distribution shift for synthetic distribution shift? **Q2**: How effective is RFR for real spatial and temporal distribution shift? **Q3**: How sensitive is RFR to the key hyperparameter $\lambda$?

### 4.1 Experimental Setting

In this subsection, we present the experimental setting, including datasets, evaluation metrics, baseline methods, distribution shift generation, and implementation details.

**Datasets.** We adopt the following datasets in our experiments. **UCI Adult** [13] dataset contains information about $45,222$ individuals with $15$ attributes from the 1994 US Census. We consider

gender as the sensitive attribute and the task is to predict whether the income of the person is higher than $50k$ or not. **ACS-I**ncome [14] is extracted from the American Community Survey (ACS) Public Use Microdata Sample (PUMS) with $3,236,107$ samples. We choose gender as the sensitive attribute. Similar to the task in UCI Adult, the task is to predict whether the individual income is above $50k$. **ACS-E**mployment [14] also derives from ACS PUMS, and we also use gender as the sensitive attribute. The task is to predict whether an individual is employed.

**Evaluation Metrics.** We use accuracy to evaluate the prediction performance for the downstream task. For fairness metrics, we adopt two common-used quantitative group fairness metrics to measure the prediction bias [15, 16], i.e., *demographic parity* $\Delta_{DP} = |\mathbb{P}(\hat{Y} = 1|A = 0) - \mathbb{P}(\hat{Y} = 1|A = 1)|$ and *equal opportunity* $\Delta_{EO} = |\mathbb{P}(\hat{Y} = 1|A = 0, Y = 1) - \mathbb{P}(\hat{Y} = 1|A = 1, Y = 1)|$, where $A$ represents sensitive attribute, $Y$ and $\hat{Y}$ represent the ground-truth label and predicted label, respectively.

**Baselines.** In our experiments, we consider vanilla multi-layer perceptron (MLP) and two widely adopted in-processing debiasing methods, including fairness regularization (REG), adversarial debiasing (ADV), and fair consistency regularization (FCR). **MLP** directly uses vanilla 3-layer MLP with 50 hidden unit and ReLU activation function [17] to minimize cross-entropy loss with source dataset. In the experiments, we adopt the same model architecture for all other methods (i.e., REG and ADV). **REG** adds a fairness-related metric as a regularization term in the objective function to mitigate the prediction bias [18, 19]. Specifically, we directly adopt demographic parity as a regularization term, and the objective function is $\mathcal{L}_{CLF} + \lambda\mathcal{L}_{DP}$, where $\mathcal{L}_{CLF}$ and $\mathcal{L}_{DP}$ are defined in Eqs. (11) and (12). **ADV** [20] employs a two-player game to mitigate bias, in which a classification network is trained to predict labels based on input features, and an adversarial network takes the output of the classification network as input and aims to identify which sensitive attribute group the sample belongs to. **FCR** [21] aims to minimize and balance consistency loss across groups.

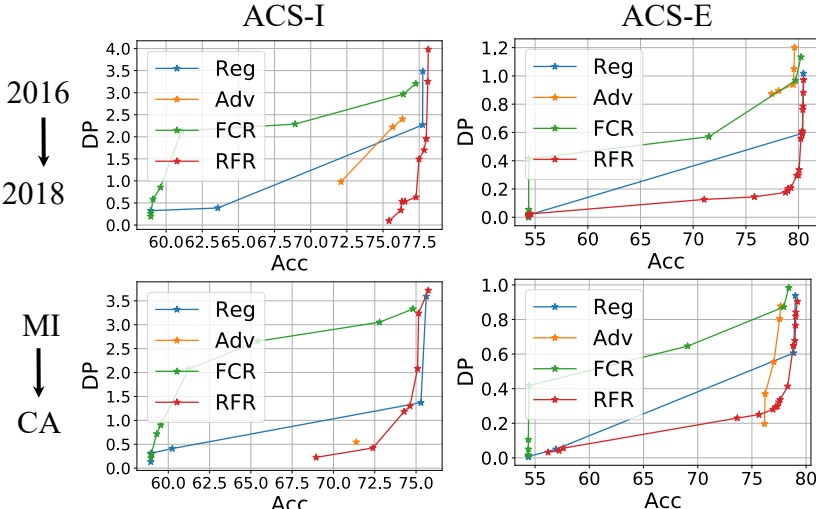

Figure 3: DP and Acc trade-off performance on three real-world datasets with temporal (Top) and spatial (Bottom) distribution shift. The trade-off curve close to the right bottom corner means better trade-off performance. The units for x- and y-axis are percentages ($\%$).

**Synthetic and Real Distribution Shift.** We adopt synthetic and real distribution shifts. For synthetic distribution shift, we follow work [22, 23] to generate distribution shift via biased sampling. Specifically, we adopt applying principal component analysis (PCA) [24] to retrieve the first principal component $\mathcal{C}$ from input attributes. Subsequently, we estimate the mean $\mu(\mathcal{C})$ and standard deviation $\sigma(\mathcal{C})$, and set Gaussian distribution $\mathcal{N}_{\mathcal{S}}(\mu(\mathcal{C}), \sigma(\mathcal{C}))$ to randomly sampling of target dataset. As for source dataset, we choose different parameters $\alpha$ and $\beta$ to generate distribution shift using another Gaussian distribution $\mathcal{N}_{\mathcal{T}}(\mu(\mathcal{C}) + \alpha, \frac{\sigma(\mathcal{C})}{\beta})$ for randomly sampling. The source and target datasets are constructed by sampling without replacement. For real distribution shift, we adopt the sampling

and pre-processing approaches following *Folktables* [14] to generate spatial and temporal distribution shift via data partition based on different US states and year from 2014 to 2018.

**Implementation Details.** We run the experiments 5 times and report the average performance for each method. We adopt Adam optimizer with $10^{-5}$ learning rate and 0.01 weight decay for all models. For baseline ADV, we alternatively train classification and adversarial networks with 70 and 30 epochs, respectively. The hyperparameters for ADV are set as $\{0.0, 1.0, 10.0, 100.0, 500.0\}$. For adding regularization, we adopt the hyperparameters set $\{0.0, 0.5, 1.0, 10.0, 30.0, 50.0\}$.

## 4.2 Experimental Results on Synthetic Distribution Shift

In this experiment, we evaluate the effectiveness of our proposed Robust Fairness Regularization (RFR) method with synthetic distribution shifts via biased sampling with different parameters $(\alpha, \beta)$. We compare the performance of RFR with several other baseline methods, including standard training, adversarial training, and fairness regularization methods. The results of performance value are presented in Table 1, and the results of fairness-accuracy tradeoff are presented in Figure 2. We have the following observations:

- The results in Table 1 demonstrate that RFR consistently outperforms the baselines in terms of fairness for small distribution shifts, achieving a better balance between fairness and accuracy. For example, RFR achieves an improvement of $47.0\%$ on metric $\Delta DP$ and $84.8\%$ on metric $\Delta EO$ compared to the second-best method in the Adult dataset with $(1.0, 2.0)$-synthetic distribution shift. Furthermore, we observed that the outperforms of RFR compared with baselines decrease as the distribution shift intensity increases. The reason is that the approximation RFR involved Taylor expansion over model perturbation and is effective with mild distribution shifts.

- The results in Figure 1 show that RFR achieved a better fairness-accuracy tradeoff compared to the baseline methods for mild distribution shift, We observed that our proposed method achieved a better Pareto frontier compared to the existing methods, and the bias can be mitigated with tiny accuracy drop.

The experimental results show that our method can effectively address the fairness problem under mild distribution shift while maintaining high accuracy, outperforming the existing state-of-the-art baseline methods.

## 4.3 Experimental Results on Real Distribution Shift

In this experiment, we evaluate the performance of RFR on multiple real-world datasets with real distribution shifts. We use ACS dataset in the experiment. The distribution of this dataset varies across different time periods or geographic locations, which also causes fairness performance degradation under the distribution shift. The results are presented in Table 2, and the results of the fairness-accuracy tradeoff are presented in Figure 3. The results show that our method consistently outperforms the baselines across all datasets, achieving a better balance between fairness and accuracy. Specifically, we have the following observations:

- Table 2 demonstrates that RFR consistently outperforms the baselines across all datasets in terms of fairness-accuracy tradeoff. This suggests that our method is effective in achieving robust fairness under real temporal and spatial distribution shifts. For example, RFR achieved an improvement of $34.9\%$ on metric $\Delta DP$ and $34.4\%$ on metric $\Delta EO$ compared to the second-best method in ACS-I dataset with temporal distribution shift.

- Figure 3 shows that RFR can achieve better fairness-accuracy tradeoff than baselines, i.e., RFR is particularly effective in addressing the fairness problem on the ACS dataset with source/target data varying across different time periods or geographic locations. This is important for many practical applications, where fairness is a critical requirement, such as in credit scoring, and loan approval.

- The variance for spatial shift on ACS-I is higher than that of temporal shift on all methods, which indicates neural networks easily converge to different local minima for spatial distribution shift. The proposed methods can achieve better fairness results for most cases and the tradeoff results clearly demonstrate the effectiveness.

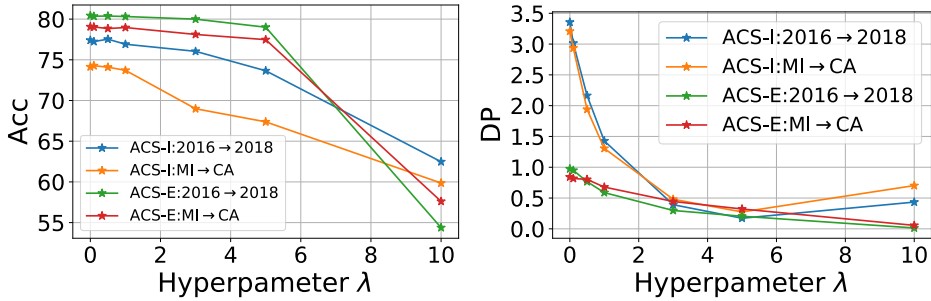

Figure 4: The hyperparameters study for $\lambda$. The **Left** subfigure and **Right** subfigure show the results of accuracy and fairness performance, respectively.

Overall, the experimental results on temporal and spatial distribution shifts further support the effectiveness of our proposed method RFR and its potential for practical applications in diverse settings.

Table 2: Performance comparison with baselines on real temporal (the year 2016 to the year 2018) and spatial (Michigan State to California State) distribution shift. The best and second-best results are highlighted with **hold** and underline, respectively.

| Real | Methods | ACS-I | | | ACS-E | | |
|---|---|---|---|---|---|---|---|
| | | Acc (%) ↑ | $\Delta_{DP}$ (%) ↓ | $\Delta_{EO}$ (%) ↓ | Acc (%) ↑ | $\Delta_{DP}$ (%) ↓ | $\Delta_{EO}$ (%) ↓ |
| $2016 \rightarrow 2018$ | MLP | 77.75±0.44 | 3.26±0.38 | 3.48±0.41 | 80.46±0.05 | 1.07±0.10 | 1.02±0.10 |
| | REG | 77.74±0.62 | 2.09±0.21 | 2.27±0.24 | 80.37±0.12 | 0.77±0.08 | 0.74±0.08 |
| | ADV | 75.94±0.40 | 2.41±0.49 | 2.53±0.55 | 79.62±0.14 | 1.17±0.14 | 1.10±0.14 |
| | FCR | 76.40±0.45 | 2.81±0.30 | 2.96±0.30 | 79.59±0.38 | 0.95±0.42 | 0.91±0.34 |
| | RFR | 77.49±0.32 | **1.36**±0.17 | **1.49**±0.17 | 80.36±0.05 | **0.61**±0.11 | **0.58**±0.10 |
| $MI \rightarrow CA$ | MLP | 75.62±0.80 | 5.22±0.86 | 3.60±0.34 | 79.02±0.20 | 0.73±0.07 | 0.94±0.05 |
| | REG | 75.52±0.78 | 2.88±0.44 | 2.17±0.22 | 75.34±1.11 | **0.42**±0.09 | **0.61**±0.11 |
| | ADV | 73.38±1.07 | **1.04**±0.58 | **0.54**±0.38 | 77.56±0.41 | 0.61±0.18 | 0.80±0.13 |
| | FCR | 74.28±0.35 | 5.06±0.62 | 3.67±0.51 | 77.96±0.22 | 0.44±0.14 | 0.67±0.38 |
| | RFR | 74.63±0.45 | 1.35±0.39 | 1.30±0.24 | 78.84±0.21 | 0.44±0.09 | 0.65±0.07 |

## 4.4 Hyperparameter Study

In this experiment, we investigate the sensitivity of the hyperparameter $\lambda$ in Equation $\mathcal{L}_{all} = \mathcal{L}_{CLF} + \lambda(\mathcal{L}_{DP} + \mathcal{L}_{RFR})$ for spatial and temporal distribution shift across different datasets. Specifically, we tune the hyperparameter as $\lambda = \{0.0, 0.1, 0.5, 1.0, 3.0, 5.0, 10.0\}$. From the results in Figure 4, it is seen that the accuracy and demographic parity are both sensitive to hyperparameter $\lambda$, which implies the capability of accuracy-fairness control. With the increase of $\lambda$, accuracy decreases while demographic parity also decreases. When $\lambda$ is smaller than 5, accuracy drops slowly while demographic parity drops faster. Such observation represents that an appropriate hyperparameter can mitigate prediction bias while preserving comparable prediction performance.

## 5 Related Work

In this section, we present two lines of related work, including fairness in machine learning and distribution shift.

**Fairness in Machine Learning.** Fairness [25]?–[31] is a legal requirement for machine learning models for various high-stake real-world predictions, such as healthcare [7, 32, 33], education [34–36], and job market [37, 38]. Achieving fairness, either from a data or model perspective [39–42], in machine learning is a challenging problem. As such, there has been an increasing interest in both the industrial and research community to develop algorithms to mitigate these issues and ensure that machine learning models make fair and unbiased decisions. Extensive efforts led to the development of various techniques and metrics for fairness and proposed various definitions of fairness, such as group fairness [43–47, 40], individual fairness [48–53], and counterfactual fairness [54–56]. In this paper, we focus on group fairness, and the widely used methods to achieve group fairness are fair regularization and adversarial debias method. [19, 18] proposed to add a fairness regularization term

to the objective function to achieve group fairness, and [20] proposes to jointly train a classification network and an adversarial network to mitigate the bias for different demographic groups to achieve group fairness. Overall, ensuring fairness in machine learning is a critical and ongoing research area that will continue to be an important focus for the development of responsible machine learning.

**Distribution Shift.** Previous work [1–4, 57] reveals that a classifier trained on a source distribution will perform worse on a given target distribution because of the distribution shift, and recently extensive works have explored the influence of distribution shift in model prediction. Moreover, distribution shift can significantly affect the fairness performance of machine learning models. The sensitivity of fairness to the distribution shift is notorious for the legal requirement. The degraded performance of fair models under a distribution shift would trigger new bias and discrimination issues. There are some works [23, 58, 21, 59, 60] that solve fairness under various distribution shifts. For example, [23] explore the fairness under covariate shift, where the inputs change with the in-distribution label. [60] proposes Shifty algorithms to hold fairness guarantees when the dataset in the deployment environment is out-of-distribution of the training datasets (distribution shift). Our work is different from those prior works from model weight perturbation perspective.

## 6    Conclusion

This paper aims the solve the fairness problem under the distribution shifts from the model weight perturbation perspective. We first establish a theoretical connection between distribution shift, data perturbation, and model weight perturbation, which allowed us to conclude that distribution shift and model perturbation are equivalent. We then propose a sufficient condition for ensuring fairness transference under distribution shift. To explicitly chase such sufficient conditions, we introduce Robust Fairness Regularization (RFR) method based on the established understanding, to achieve robust fairness. Our experiments on both synthetic and real distribution shifts demonstrate the effectiveness of RFR in achieving a better fairness-accuracy tradeoff compared to existing baselines. We believe that our understanding of distribution shift is valuable and intriguing to the development of robust machine learning models, and the proposed RFR approach can be of great practical value to build fair and robust machine learning models in real-world applications.

## 7    Acknowledgement

The authors thank the anonymous reviewers for their helpful comments. The work is in part supported by NSF grants NSF IIS-2224843, IIS-1939716, and IIS-1900990. The views and conclusions contained in this paper are those of the authors and should not be interpreted as representing any funding agencies.

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
