}})$ can move source distribution $P_{\mathcal{S}}$ to target distribution $P_{\mathcal{T}}$. Define data perturbation $\delta = T - S$ and joint distribution of $S$ and $T$ is given by $\gamma^*(s, t)$. We can obtain the following probability for any $u$

$$F(u) = \mathbb{P}(\delta \leq u) = \int_{\mathcal{S}} \int_{\mathcal{T}_{s,u}} \gamma^*(s, t) \mathrm{d}t \mathrm{d}s, \tag{14}$$

where $\mathcal{T}_{s,u} = \{t : t \leq u + s\}$. The probability of data perturbation $\delta$ is given by

$$\mathcal{P}(\delta) = \frac{\partial F(u)}{\partial u}\Big\|_{u=\delta} = \int_{\mathcal{S}} \gamma^*(s, s + \delta) \mathrm{d}s, \tag{15}$$

Note that, for any positive $p > 0$, the power of data perturbation $\delta$ satisfy

$$\mathbb{E}[||\delta||_p^p] = \mathbb{E}[||T - S||_p^p] = \iint ||s - t||_p^p \gamma^*(s, t) \mathrm{d}s \mathrm{d}t, \tag{16}$$

where $|| \cdot ||_p$ represents $L_p$ norm. The data perturbation $\delta$ based on Eq. (1) with square cost function $c(s, t) = ||s - t||_p^p$ has minimal power.

## B  Proof of Corollary 2.2

Based on Theorem 2.1, it is easy to see $(X + \delta_X(X), Y + \delta_Y(Y)) \sim P_{\mathcal{T}}$ if $(X, Y) \sim P_{\mathcal{S}}$, therefore, the following equality holds for any loss function $l(\cdot, \cdot)$:

$$\mathbb{E}_{(X,Y) \sim P_{\mathcal{T}}}[l(f_\theta(X), Y)]$$
$$= \mathbb{E}_{\delta_X(X), \delta_Y(Y)} \mathbb{E}_{(X,Y) \sim P_{\mathcal{S}}}[l(f_\theta(X + \delta_X(X)), Y + \delta_Y(Y))].$$

## C  Proof of Theorem 2.3

We consider the small distribution shift, and smooth training loss and model prediction function, the equivalent data perturbation is also small. We conduct first-order Taylor expansion over loss function $l(f_\theta(X + \delta_X(X)), Y + \delta_Y(Y))$, we have

$$l(f_\theta(X + \delta_X(X)), Y + \delta_Y(Y)) \tag{17}$$
$$= l(f_\theta(X), Y) + \frac{\partial l}{\partial f} \frac{\partial f}{\partial X}\Big|_{X=\hat{X}} \delta_X(X) + \frac{\partial l}{\partial f} \frac{\partial f}{\partial Y}\Big|_{Y=\hat{Y}} \delta_Y(Y),$$

where $\hat{X}$ is between $X$ and $X + \delta_X(X)$, and $\hat{Y}$ is between $Y$ and $Y + \delta_Y(Y)$. For simplicity, we ignore $\hat{X}$ and $\hat{Y}$ in the following derivation. Subsequently, we can approximate the training loss on the target dataset as follows:

$$\mathbb{E}_{\delta_X(X), \delta_Y(Y)} \mathbb{E}_{(X,Y) \sim P_{\mathcal{S}}}[l(f_\theta(X + \delta_X(X)), Y + \delta_Y(Y))]$$
$$= \mathcal{R}_{\mathcal{S}} + \mathbb{E}_{\delta_X(X), \delta_Y(Y)} \mathbb{E}_{(X,Y) \sim P_{\mathcal{S}}}[\frac{\partial l}{\partial f} \frac{\partial f}{\partial X} \delta_X(X) + \frac{\partial l}{\partial f} \frac{\partial f}{\partial Y} \delta_Y(Y)]$$
$$= \mathcal{R}_{\mathcal{S}} + \mathbb{E}_{(X,Y) \sim P_{\mathcal{S}}}[\frac{\partial l}{\partial f} \frac{\partial f}{\partial X}] \cdot \mathbb{E}_{\delta_X(X)}[\delta_X(X)]$$
$$+ \mathbb{E}_{(X,Y) \sim P_{\mathcal{S}}}[\frac{\partial l}{\partial f} \frac{\partial f}{\partial Y}] \cdot \mathbb{E}_{\delta_Y(Y)}[\delta_Y(Y)]. \tag{18}$$

As for model weight perturbation $\Delta\theta$, according to first-order Taylor expansion, we have

$$l(f_{\theta+\Delta\theta}(X), Y)] = l(f_\theta(X), Y) + \frac{\partial l}{\partial f} \frac{\partial f}{\partial \theta}\Big|_{\theta=\hat{\theta}} \Delta\theta, \tag{19}$$

where $\hat{\theta}$ is between $\theta$ and $\theta + \Delta\theta$. Subsequently, we can approximate the training loss on the source dataset as follows:

$$\mathbb{E}_{(X,Y)\sim\mathcal{P}_\mathcal{S}}[l(f_{\theta+\Delta\theta}(X),Y)]$$
$$= \mathcal{R}_\mathcal{S} + \mathbb{E}_{(X,Y)\sim\mathcal{P}_\mathcal{S}}\Big[\frac{\partial l}{\partial f}\frac{\partial f}{\partial \theta}\Big]\Big|_{\theta=\hat{\theta}}\Delta\theta, \tag{20}$$

Compared Eqs. (18) and (20), for any data perturbation, the model weight perturbation is treated as multivariate but with only one equation. In other words, considering linear equation $\mathbf{A}\mathbf{x} = b$, where $\mathbf{A} \in \mathbb{R}^{1\times n}$ and $b \in \mathbb{R}^{1\times 1}$ are both constant, $\mathbf{x} \in \mathbb{R}^{n\times 1}$ is variables, the goal is to find whether the solution $\mathbf{x}$ exists for linear equation $\mathbf{A}\mathbf{x} = b$. Note that we consider distribution shift problem, i.e., $b \neq 0$, the solution for linear equation $\mathbf{A}\mathbf{x} = \mathbf{b}$ exists when $rank([A|b]) = 1 = rank(A)$, i.e., $\|A\| = \Big\|E_{(X,Y)\sim\mathcal{P}_\mathcal{S}}[\frac{\partial l}{\partial f}\frac{\partial f}{\partial \theta}]\big|_{\theta=\hat{\theta}}\Big\| > 0$. Note that such a non-zero gradient condition is easily satisfied for models that are not well-trained. Therefore, we can always find a model weight perturbation so that the training loss on source dataset with data perturbation $\delta$ and model weight perturbation $\Delta\theta$ are the same.

## D More details on Eq. (5)

**Lemma D.1.** *For any scale $a_1$, $a_2$, $b_1$, and $b_2$, we have*
$\big||a_1 - b_1| - |a_2 - b_2|\big| \leq |a_1 - a_2| + |b_1 - b_2|$.

For any scale $a_1$, $a_2$, $b_1$, and $b_2$, it is easy to check that

$$\big||a_1 - b_1| - |a_2 - b_2|\big| \leq |a_1 - a_2| + |b_1 - b_2|$$
$$\Longleftrightarrow \quad -2a_1b_1 - 2a_2b_2 - 2|a_1 - b_1||a_2 - b_2|$$
$$\leq -2a_1a_2 - 2b_1b_2 + 2|a_1 - a_2||b_1 - b_2|$$
$$\Longleftrightarrow \quad |a_1 - a_2||b_1 - b_2| + |a_1 - b_1||a_2 - b_2|$$
$$+a_1b_1 + a_2b_2 - a_1a_2 - b_1b_2 \geq 0, \tag{21}$$

Notice that

$$-(a_1 - a_2)(b_1 - b_2) + (a_1 - b_1)(a_2 - b_2)$$
$$+a_1b_1 + a_2b_2 - a_1a_2 - b_1b_2 = 0, \tag{22}$$

Eq. (21) holds, and the proof is completed.

## E More details on Training Acceleration

Considering that the proposed RFR is computation-expensive due to the inherent maximization problem, it is intractable to adopt RFR during model training. To this end, we develop an efficient and effective approximation to model weight perturbation for the worst case and the gradient of RFR. In this way, the optimizer (e.g., stochastic gradient descent) can be directly adopted for training. Specifically, we first approximate the maximization problem via a first-order Taylor expansion. For $\mathcal{L}_{RFR,\mathcal{S}_0}$, the optimal model weight perturbation is given by

$$\epsilon_0^*(\theta) = \arg\max_{\|\epsilon_0\|_p\leq\rho} \mathbb{E}_{\mathcal{S}_0}[f_{\theta+\epsilon_0}(\mathbf{x})] - \mathbb{E}_{\mathcal{S}_0}[f_\theta(\mathbf{x})]$$
$$\approx \arg\max_{\|\epsilon_0\|_p\leq\rho} \frac{\partial\mathbb{E}_{\mathcal{S}_0}[f_\theta(\mathbf{x})]}{\partial\theta}\epsilon_0 \overset{\triangle}{=} \arg\max_{\|\epsilon_0\|_p\leq\rho} g_0\epsilon_0,$$

where $g_0 = \frac{\partial\mathbb{E}_{\mathcal{S}_0}[f_\theta(\mathbf{x})]}{\partial\theta}$ represents the gradient of average prediction for source data with sensitive attribute $A = 0$. Note that the number of model parameters is usually high, the higher order terms computation is time-consuming. For example, the second term involves Hessian matrix with square polynomial complexity. Therefore, we omit high-order terms and only keep one order term to accelerate training. In turn, the optimal model weight perturbation is given by the solution of a classical dual norm problem, i.e.,

$$\epsilon_0^*(\theta) = \rho \cdot \text{sign}(g_0)\frac{|g_0|^{q-1}}{\big(\|g_0\|_q^q\big)^{1/p}}, \tag{23}$$

where $\frac{1}{p} + \frac{1}{q} = 1$, $\text{sign}(\cdot)$ is element-wise sign function, and $|\cdot|^{q-1}$ denotes element-wise absolute value and power. Considering gradient-based optimizer for model training, the gradient for $\mathcal{L}_{RFR,\mathcal{S}_0}$ is given by

$$
\begin{aligned}
\nabla_\theta \mathcal{L}_{RFR,\mathcal{S}_0} &\approx \frac{\partial \mathbb{E}_{\mathcal{S}_0}[f_{\theta+\epsilon_0^*(\theta)}(\mathbf{x})]}{\partial \theta} \\
&= \frac{\partial \mathbb{E}_{\mathcal{S}_0}[f_\theta(\mathbf{x})]}{\partial \theta} + \frac{\partial \epsilon_0^*(\theta)}{\partial \theta} \frac{\partial \mathbb{E}_{\mathcal{S}_0}[f_\theta(\mathbf{x})]}{\partial \theta}.
\end{aligned}
\tag{24}
$$

It is seen that the approximation of $\nabla_\theta \mathcal{L}_{RFR,\mathcal{S}_0}$ can be directly calculated via automatic differentiation. However, the calculation of the term $\frac{\partial \epsilon_0^*(\theta)}{\partial \theta}$ implicitly depends on the Hessian of $\mathbb{E}_{\mathcal{S}_0}[f_\theta(\mathbf{x})]$ due to $\epsilon_0^*(\theta)$ is a function of $g_0$. To further accelerate the computation, we drop the second-order term and the final approximation of $\nabla_\theta \mathcal{L}_{RFR,\mathcal{S}_0}$ is given by

$$
\nabla_\theta \mathcal{L}_{RFR,\mathcal{S}_0} \approx \frac{\partial \mathbb{E}_{\mathcal{S}_0}[f_\theta(\mathbf{x})]}{\partial \theta}\Big|_{\theta+\epsilon_0^*(\theta)},
\tag{25}
$$

where $\epsilon_0^*(\theta)$ is given by Eq. (23). Similarly, we can obtain the approximation of $\nabla_\theta \mathcal{L}_{RFR,\mathcal{S}_1}$ as follows:

$$
\nabla_\theta \mathcal{L}_{RFR,\mathcal{S}_1} \approx \frac{\partial \mathbb{E}_{\mathcal{S}_1}[f_\theta(\mathbf{x})]}{\partial \theta}\Big|_{\theta+\epsilon_1^*(\theta)},
\tag{26}
$$

where $g_1 = \frac{\partial \mathbb{E}_{\mathcal{S}_1}[f_\theta(\mathbf{x})]}{\partial \theta}$ and model weight perturbation for group $A = 1$ is given by

$$
\epsilon_1^*(\theta) = \rho \cdot \text{sign}(g_1) \frac{|g_1|^{q-1}}{\left(\|g_1\|_q^q\right)^{1/p}}.
\tag{27}
$$

## F More Experimental Results

We provide more experimental results to further support the effectiveness of our proposed RFR.

### F.1 More Experimental Results on Training Datasets

In this experiment, we evaluate the fairness-accuracy performance, as shown in Figure 5, for source and target datasets on two real-world datasets with temporal and spatial distribution shifts. We observe that RFR achieves a better fairness-accuracy tradeoff on the target dataset compared to the baseline methods for temporal and spatial distribution shifts, while RFR does not always perform best on the source dataset in terms of fairness-accuracy tradeoff.

### F.2 More Experimental Results on EO

We also report the acc-EO tradeoff performance for two real-world datasets compared with many baselines in Figure 6. It is seen that similar to acc-DP tradeoff performance, the tradeoff performance of RFR is the best among all baselines. Please notice that there is no revision for REG and ADV methods, i.e., REG and ADV are designed for DP.

### F.3 Experimental Results without Distribution Shifts

To further investigate the effect of distribution shifts, we also provide the experimental results without distribution shifts. Specifically, we avoid the data partition and randomly split the data samples across multiple years and multiple states in two real-world datasets. The fairness and accuracy and the corresponding results are shown in Table 3 and Figure 5. We have the following observations:

- Table 3 demonstrate that REG and ADV serve as strong baselines for this setting and our method can only achieve the best results in ACS-I dataset. This suggests that the applicable scope of our method is limited. For example, for the setting without distribution shifts, the conventional methods, such as REG and ADV, may perform better. Such observation further validates the effectiveness of our proposed RFR in tackling the distribution shifts problem.

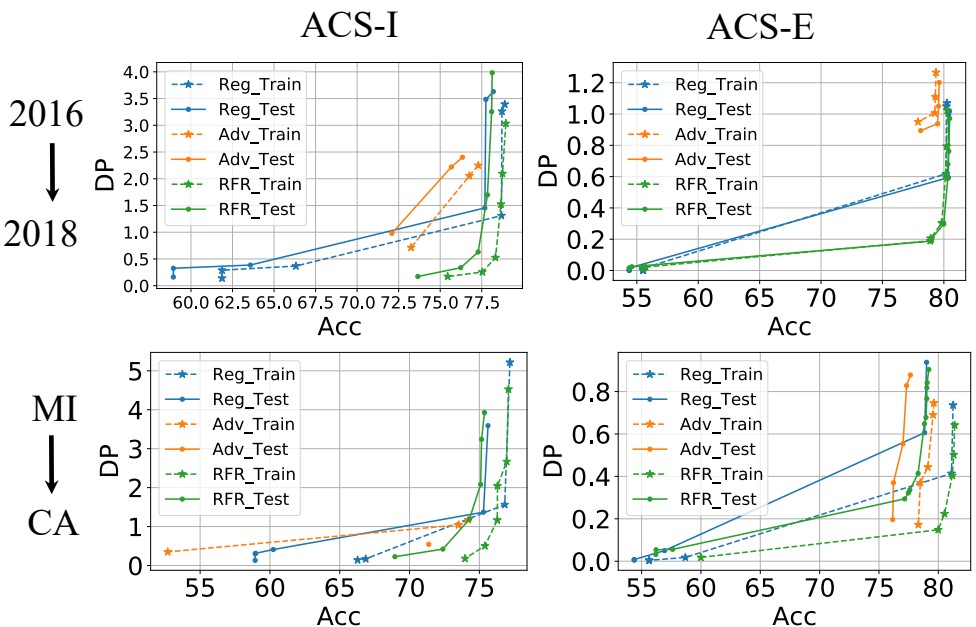

Figure 5: DP and Acc trade-off performance on three real-world datasets with temporal (Top) and spatial (Bottom) distribution shifts for source (train) and target (test) datasets. The trade-off curve close to the right bottom corner means better trade-off performance. The units for x- and y-axis are percentages (%).

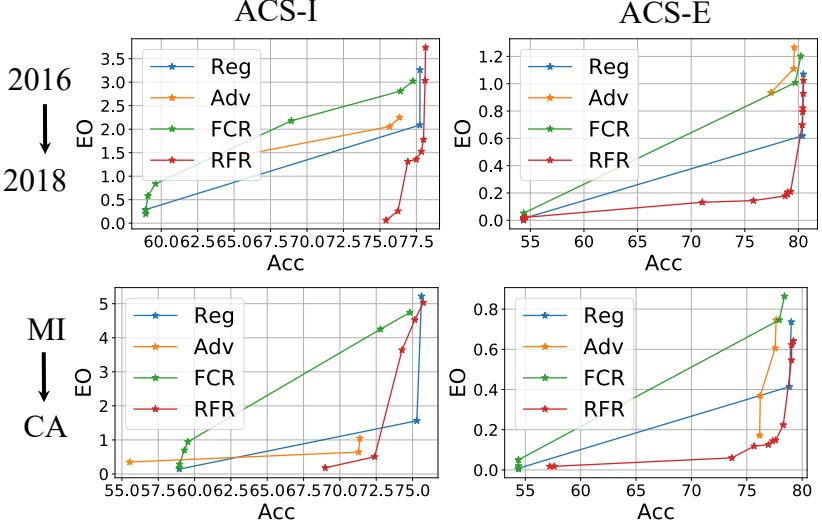

Figure 6: EO and Acc trade-off performance on two real-world datasets without distribution shift.

- Figure 5 shows that REG and ADV are comparable with our proposed RFR in terms of accuracy and fairness tradeoff performance. This observation implies the importance of distribution shift intensity identification, which will serve as important prior knowledge for algorithm or model selection.

## F.4 Hyperparameter study on $\rho$

We conduct the hyperparameter study on $\rho$ to validate the effectiveness of our proposed RFR considering the worst of weight perturbation, as shown in Figure 8. We have the following observations:

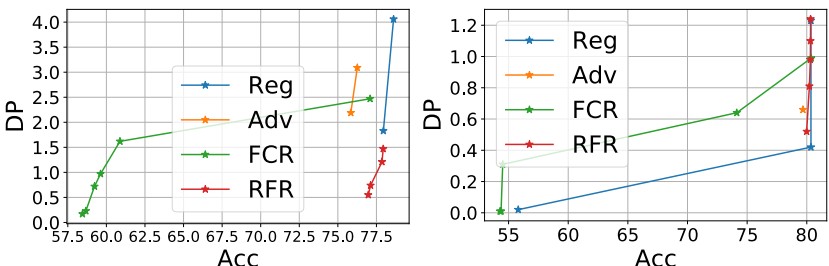

Figure 7: DP and Acc trade-off performance on two real-world datasets without distribution shift. The trade-off curve close to the right bottom corner means better trade-off performance.

Table 3: Performance comparison with baselines without distribution shift. The best and second-best results are highlighted with **hold** and underline, respectively.

| Methods | ACS-I | | | ACS-E | | |
|---|---|---|---|---|---|---|
| | Acc (%) ↑ | $\Delta_{DP}$ (%) ↓ | $\Delta_{EO}$ (%) ↓ | Acc (%) ↑ | $\Delta_{DP}$ (%) ↓ | $\Delta_{EO}$ (%) ↓ |
| MLP | 78.61±0.42 | 3.82±0.21 | 4.06±0.43 | 80.29±0.08 | 1.23±0.18 | 0.89±0.21 |
| REG | 77.82±0.51 | 2.66±0.28 | 2.61±0.34 | 80.35±0.17 | **1.05**±0.13 | 1.23±0.16 |
| ADV | 75.85±0.62 | 2.13±0.59 | 1.90±0.45 | 79.48±0.24 | 1.15±0.18 | **0.59**±0.12 |
| FCR | 75.88±0.81 | 2.60±0.31 | 2.94±0.33 | 79.90±0.15 | 1.07±0.07 | 1.18±0.06 |
| RFR | 77.57±0.51 | **1.88**±0.36 | **1.40**±0.36 | 80.12±0.13 | 1.09±0.07 | 0.70±0.07 |

- The accuracy and DP metrics are both sensitive to hyperparameter $\rho$, where ACS-I dataset is even more sensitive to ACS-E. In other words, the optimal hyperparameter $\rho$ to achieve robust fairness while preserving accuracy is dependent on the dataset and distribution shifts intensity, which implies that there are opportunities to further improve the tradeoff performance by hyperparameter selection.

- It is not necessarily that a large perturbation ball leads to a better fairness performance. The reasons are two-fold. Firstly, we only consider the worst case of weight perturbation, which is not consistent with real distribution shifts. Secondly, the optimal (worst case) perturbation vector is hard to find in practice. In the implementation of RFR, we use the Taler expansion to approximately find the weight perturbation for training acceleration in Appendix E.

### F.5 Ablation study on $\mathcal{L}_{DP}$ for different hyperparameter $\lambda$

We conduct the aablation study on $\mathcal{L}_{DP}$ to validate the effectiveness of our proposed RFR as shown in Figure 9. We have the following observations:

- $\mathcal{L}_{DP}$ loss term is very important to achieve robust fairness under distribution shift. Without $\mathcal{L}_{DP}$ loss, the DP cannot be mitigated significantly with a large hyperparameter. Such observation is consistent with Eq. (5).

- ACS-I dataset is more sensitive to hyperparameter $\lambda$ compared with ACS-E. This suggests the necessity of tuning the hyperparameter carefully for each dataset.

## G RFR Algorithms

The pseudo-code of RFR algorithm is given by Algorithm 1. The two forward and two backward propagations happen in Lines 6 and 7.

## H More Discussion

### H.1 Discussion on Decision Tree Extention

Our method is designed for neural networks and can not be used for decision tree methods (e.g., XG-Boost, GBDT). The key reason is that our algorithm (See Appendix G) involves gradient computation

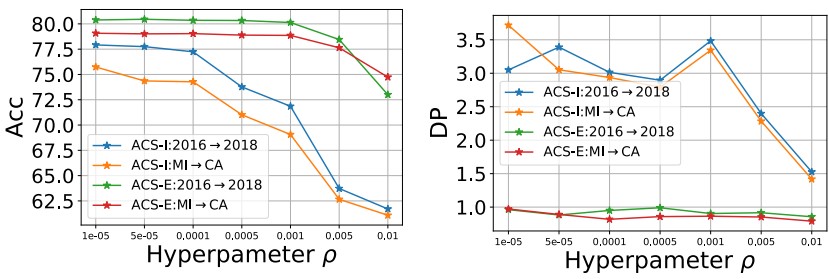

Figure 8: Hyperparameter study on $\rho$.

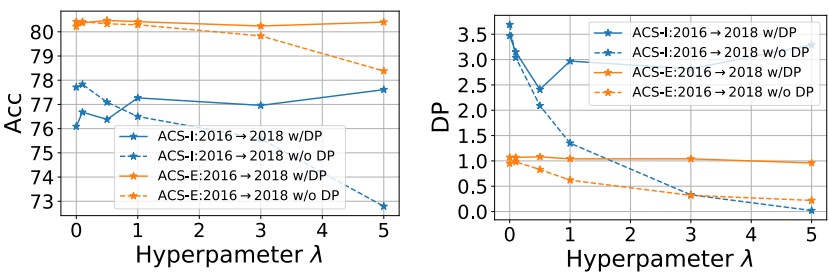

Figure 9: Ablation study on $\mathcal{L}_{DP}$ for different hyperparameter $\lambda$.

and weight perturbation to accelerate the bi-level optimization problem-solving. We summarize the details as follows:

- Nature of Parameters: In a neural network, the parameters are continuous values (weights, biases) that are updated using gradients to minimize the loss. In GBDT or XGBoost, the "parameters" are the structure of the trees themselves, including the split points and leaf values. These are not continuous values that can be updated with gradient descent in the traditional sense.

- Training Mechanism: While GBDT and XGBoost use the concept of gradients (specifically, gradient boosting works by fitting new trees to the negative gradient of the loss), this is not the same as computing gradients with respect to weights in a neural network and updating them. Instead, trees are added to the model to correct the errors (residuals) of the current ensemble.

- Non-Differentiability: Decision trees involve making decisions based on hard thresholds, which are inherently non-differentiable operations. This makes them unsuitable for traditional gradient-based optimization.

Therefore, we leave the extension of this work for decision tree-based models in future work.

## H.2 Discussion Related to Existing Work

The approach of using the worst-case bound as a regularizer has been explored before for selection bias [61]. The difference is two-fold: (1) The problem setting. Work [61] considers the fairness problem with selection bias, where the selection bias is described with available auxiliary information. Our paper mainly focuses on the fairness problem under distribution shift without any information on target distribution, which can be but is not necessarily caused by selection bias. (2) Methodology. Work [61] mainly focuses on Consistent Range Approximation (CRA) of a fairness query using probability information in data collection. Our paper mainly focuses on DP metric difference in source and target distribution, and then further derives a model perturbation approach to achieve fairness under distribution.

Additionally, Sharpness-Aware Minimization (SAM) [12, 62, 63] aims to encourage the training to converge to a flatter region in which the training losses in the neighborhood around the minimizer are lower. In this paper, we mainly focus on the fairness performance under distribution shift while SAM

---
**Algorithm 1** Robust Fairness Regularization (RFR)
---
1: **Input:** Training (source) dataset $\mathcal{S} = \cup_{i=1}^{N}\{(x_i, y_i, a_i)\}$, hyperparameters $\lambda$, $\rho$, $p$.
2: **Output:** Robust fair model $f_\theta(\cdot)$.
3: Initialize model weight $\theta_0$, step size $\eta$, update step $t=0$.
4: **while** not convergence **do**
5:     Compute gradient for $\nabla_\theta \mathcal{L}_{CLF} + \lambda \nabla_\theta \mathcal{L}_{DP}$.
6:     Calculate model weight perturbation $\epsilon_0^*$ and $\epsilon_1^*$ based on Eqs. (23) and (27).
7:     Approximate gradient of RFR $\nabla_\theta \mathcal{L}_{RFR}$ based on Eqs (24) and (26).
8:     Update model weights: $\theta_{t+1} = \theta_t - \eta(\nabla_\theta \mathcal{L}_{CLF} + \lambda \nabla_\theta \mathcal{L}_{DP}) + \lambda \nabla_\theta \mathcal{L}_{RFR}$.
9:     $t = t + 1$.
10: **end while**
---

improves the generalization performance. Techniquely, the objectives and the considered sample batch of RFR are different from that of SAM.

### H.3 Future Work

There are several future directions: (1) The proposed method is specifically developed for demographic parity. As for the robust fairness for other metrics, such as counterfactual fairness, individual fairness, and other group fairness, we leave these extension works in future work. (2) We mainly focus on model weight perturbation to achieve robust fairness. Another possible approach is the input perturbation method, which is complicated since the input perturbation is dependent on input samples. Additionally, input perturbation is highly related to feature type (numerical, categorial, or mixed). (3) Note that it is intractable to select the optimal without accessing target data distribution. One promising future direction is to access more information (e.g., few-shot target samples or input features in the target dataset), the hyperparameter tuning can be done and achieve better performance . For example, if the input features in (unlabeled) target dataset are available, it is tractable to predict out-of-distribution error and then use it for hyperparameter selection [64, 65]. Note that the input feature in the target dataset should be available in the inference stage.

## I  Broader Impact

Algorithmic Fairness focuses on ensuring fairness and lack of bias in the decisions made by algorithms or machine learning models. Fairness in socio-technical systems goes beyond algorithmic fairness, considering broader societal impacts, ethical considerations, data biases, and interdisciplinary collaboration. It focuses on human-centered design, policy changes, and ongoing assessment to ensure technology aligns with societal values and promotes equitable outcomes.