# OpenReview forum: "Chasing Fairness Under Distribution Shift: A Model Weight Perturbation Approach"
_NeurIPS.cc/2023/Conference — NeurIPS 2023 poster_

### Official Review · Reviewer_3kzb · 2023-06-26

**Soundness:** 3 good
**Presentation:** 3 good
**Contribution:** 3 good
**Rating:** 7
**Confidence:** 4

**Summary:**

This paper tackles the issue of fairness in situations where there are distribution shifts, viewing it from the angle of model weight perturbation. The authors establish a theoretical link between distribution shift, data perturbation, and model weight perturbation, leading to the conclusion that distribution shift and model perturbation are essentially the same. They propose a sufficient condition to ensure fairness transference under distribution shift and introduce the Robust Fairness Regularization (RFR) method to meet these conditions and achieve robust fairness. Experimental results on both synthetic and real distribution shifts show the RFR method outperforms existing baselines in terms of fairness-accuracy tradeoff.

**Strengths:**

1. The fairness under covariate shift is an important problem. The authors also manage to show that Data Perturbation equals Model Weight Perturbation and use this motivation to construct the following methodology descriptions and theoretical analysis. The motivation and writing is strong.
2. The method proposed by this work is novel about the insights from optimal transport to optimize the worst case within the model weight perturbation ball and Data Perturbation equals Model Weight Perturbation. This paper points out the limitations of previous works well.
3. The experiments on fairness accuracy tradeoff and hyperparameter study are sufficient. Although in the table it does not explicitly outperforms baselines, the tradeoff is clearly better.

**Weaknesses:**

I do not find important weaknesses on this paper.

**Questions:**

The hyperparameter tuning seems important. Do you record the experimental setting variables and preferably provide means to reimplement the result?

----------
I have seen other reviews and am a little confused and quite frankly appalled why we need to assume zero access to data from the target distribution. Is it even possible if we do not even know data from the target distribution (or the distribution variation) to do transfer learning? Although it would be interesting to know a good $\rho$ a priori by either prior knowledge or theoretical bounds, many papers out there also do not discuss hyperparameters like $\rho$, and they are all fine. I fail to see why this could be a MAJOR problem. In my view, it is not even a weakness, but something that would simply be good to know and good to ask the authors about. I would like to raise my confidence to 4 and advocate the paper if needed.

**Limitations:**

The authors do not talk about limitations of this paper.

---

> ### Author Rebuttal · Authors · 2023-08-08
>
> Thank you for spending time and effort reviewing our paper. We appreciate your constructive comments and support to our paper. This feedback truly encouraged us to redouble our efforts in conducting quality and impactful research.
>
>
> **[Q1] The hyperparameter tuning seems important. Do you record the experimental setting variables and preferably provide means to reimplement the result?**
>
> **Ans:** We thank the reviewer for this constructive comment. The experiment result of tuning the key hyper-parameter $\rho$ of RFR has been given in Figure 1 of the one-page rebuttal. It is observed that the accuracy-fairness trade-off of RFR is stable for $1\times 10^{-5} \leq \rho\leq 1\times 10^{-3}$. This significantly indicates RFR is simple and easy to implement. Other hyper-parameter settings of the experiment have been given in Section 4.1.5 *Implementation Details* for the readers to reproduce our results.
>
> **[Q2] I have seen other reviews and am a little confused and quite frankly appalled why we need to assume zero access to data from the target distribution. Is it even possible if we do not even know data from the target distribution (or the distribution variation) to do transfer learning?**
>
> **Ans:** Yes. We target the fairness problem without accessing target distribution in this paper. Our proposed method RFR can mitigate the unfairness under distribution shift without the knowledge of target distribution. As shown in Table 1 of our paper, the experiment on three datasets comprehensively demonstrates the effectiveness of RFR, where RFR achieves a better accuracy-fairness tradeoff than the four baseline methods. Moreover, the results of a hyper-parameter study in Figure 4 of our paper show that RFR is stable towards the value of $\rho$, indicating that RFR is easy to implement and re-produce in practical scenarios.

---

> > ### Comment · Reviewer_3kzb · 2023-08-12
> >
> > I have read the rebuttal. I remain my score.

---

> > > ### Author Response · Authors · 2023-08-14
> > > **Thanks for your quick response.**
> > >
> > > Thanks for your quick response and support for our paper.

---

### Official Review · Reviewer_bBot · 2023-07-03

**Soundness:** 3 good
**Presentation:** 4 excellent
**Contribution:** 3 good
**Rating:** 6
**Confidence:** 2

**Summary:**

The paper is in the context of fairness measures under distribution shift.
There is first a theoretical connection between distribution shift, data perturbation and model perturbations.
The proposed method Robust Fairness Regularizatin (RFR), proposes a regularization using 3 terms L_CLF-- classifier loss function, L_DP ensuring DP on source data and L_RFR= L_RFR_S0 +  L_RFR_S1 which is the fairness transferability from training set.
The experiments are in  synthetic (PCA transformations) and real data (Adult and Folk)

**Strengths:**

The paper is extremely well-written and clear. Enjoyable lecture.

The method proposed is simple.

The work seems extensive and thorough.

Experiments are well-redacted and easy to understand.

The work is one of the few (to the best of my knowledge) that are tackling distribution shift and fairness, without strong assumptions like causal graph availability.

**Weaknesses:**

The method is designed/tested in Neural Network architectures in tabular datasets. In tabular data algorithms like gradient boosting decision trees achieve SOTA[1] and are restricted to Linear Regression in regulated environments.



The paper calls fairness to just one metric --Demographic Parity-- Fairness is not just one metric. L_RFR is not performance based but group disparate treatment based.

The work does not acknowledge the difference between algorithmic fairness and fairness in broader socio-technical systems.



## Minor

L72 Citation [9] is a specific "fairness distribution shift" citation, not a general one about distribution shift, which is what is about the sentence.

I wonder how different are actually Adult and ACS Income.

There seem to be many citations coming from Arxiv. Perhaps it could be good to check if there are more recent proceedings versions.


[1] https://proceedings.neurips.cc/paper_files/paper/2022/file/0378c7692da36807bdec87ab043cdadc-Paper-Datasets_and_Benchmarks.pdf

**Questions:**

The work states that "any distribution shift can be equivalent to data perturbations." Does this approach (Sec2.2) considers pure concept shift? When $P(X_S)=P(X_T)$ and $P(Y_S)=P(Y_T)$, but the only thing that changes is the relationship between covariate and target? $X\Rightarrow Y$.

Is the method strictly limited to NN? If so (since its tabular data) it may be worth stating it earlier and clearer in the paper.
I wonder if the theoretical results from section 2 can be aligned with the previous empirical findings of https://arxiv.org/pdf/2302.05018.pdf

**Limitations:**

Fairness is just one metric. I wonder if this limitation should be stated early on the paper or even affect the title.

The work does not acknowledge the difference between algorithmic fairness and fairness in broader socio-technical systems.

---

> ### Author Rebuttal · Authors · 2023-08-08
>
> We thank the reviewer for the constructive comments.
>
> **[W1] The method is designed/tested in Neural Network architectures in tabular datasets. In tabular data algorithms like gradient boosting decision trees achieve SOTA[1] and are restricted to Linear Regression in regulated environments.**
>
> **Ans:** In this paper, we only consider the classification task and we leave the regression task in future work. We will add more discussion on the benchmark paper [1] in our revised version.
>
>
> **[W2] The paper calls fairness to just one metric --Demographic Parity-- Fairness is not just one metric. L_RFR is not performance-based but group disparate treatment based.**
>
> **Ans:** $L_{RFR}$ is specifically designed for demographic parity. There are many other fairness metrics, such as equal opportunities, and individual fairness, that could be specifically designed under distribution shift. We leave these parts in the future work. We will add the discussion in Appendix~H.
>
> **[W3] The work does not acknowledge the difference between algorithmic fairness and fairness in broader socio-technical systems.**
>
> **Ans:** We will add acknowledge the difference between algorithmic fairness and fairness in broader socio-technical systems. Algorithmic Fairness focuses on ensuring fairness and lack of bias in the decisions made by algorithms or machine learning models. Fairness in socio-technical systems goes beyond algorithmic fairness, considering broader societal impacts, ethical considerations, data biases, and interdisciplinary collaboration. It focuses on human-centered design, policy changes, and ongoing assessment to ensure technology aligns with societal values and promotes equitable outcomes.
>
> **[Q1] L72 Citation [9] is a specific "fairness distribution shift" citation, not a general one about distribution shift, which is what is about the sentence.**
>
> **Ans:** We thank the reviewer for this comment. We would like to clarify that our work targets the fairness problem under distribution shift. We believe the initial reference [9] can provide comprehensive background information and motivation for supporting the necessity of our work. However, from the perceptive of the reviewer, we concur that it is better to cite another work about general distribution shift here. Therefore, we will also cite existing work [2] on the general distribution shift here in the camera-ready version.
>
> [2] Wiles, Olivia, et al. "A Fine-Grained Analysis on Distribution Shift." International Conference on Learning Representations. 2022.
>
> **[Q2] I wonder how different are actually Adult and ACS Income.**
>
> **Ans:** ACS Income dataset is coming from paper [3]. [3] releases a more comprehensive Adult dataset, which is a superset of the UCI Adult data from available US Census sources. The data span multiple years and all states of the United States, allowing researchers to study temporal and spatial distribution shifts.
>
> [3] Ding, Frances, et al. "Retiring adult: New datasets for fair machine learning." Advances in neural information processing systems 34 (2021): 6478-6490.
>
> **[Q3] There seem to be many citations coming from Arxiv. Perhaps it could be good to check if there are more recent proceedings versions.[1] https://proceedings.neurips.cc/paper_files/paper/2022/file/0378c7692da36807bdec87ab043cdadc-Paper-Datasets_and_Benchmarks.pdf**
>
> **Ans:** We thank the reviewer for this thoughtful comment. As the reviewers mentioned, we will revise the Arxiv reference into proceeding versions in the camera-ready version.
>
>
> **[Q4] The work states that "any distribution shift can be equivalent to data perturbations." Does this approach (Sec2.2) considers pure concept shift? When $P(\mathbf{X}_s)=P(\mathbf{X}_T)$ and $P(\mathbf{Y}_s)=P(\mathbf{Y}_T)$
> , but the only thing that changes is the relationship between covariate and target?  $X \Rightarrow Y$.**
>
> **Ans:** We consider a general distribution shift case. We would like to highlight that the data perturbation $\delta_X$ and $\delta_Y$ are related to $X$ and $Y$. The pure concept shift can be included in data perturbation. For example, considering two examples $(X_1, Y_1)$ and $(X_2, Y_2)$, we can only add feature data perturbation to exchange features as $(X_2, Y_1)$ and $(X_1, Y_2)$. In this way, the data marginal distribution keeps the same but the relation between features and labels could be changed.
>
> **[Q5] Is the method strictly limited to NN? If so (since it's tabular data) it may be worth stating it earlier and clearer in the paper. I wonder if the theoretical results from section 2 can be aligned with the previous empirical findings of [4] https://arxiv.org/pdf/2302.05018.pdf.**
>
> **Ans:** We will mention that our paper is designed for NN and mentioned earlier in the paper. The extension for other models, such as decision tree, would be an interesting direction in the future, especially for SOTA performance of decision tree in regression tasks. As for comparison with paper [4], [4] adopts optimal transport to predict model performance while we use optimal transport to understand distribution shift problem and derive the model perturbation problem.

---

> > ### Comment · Reviewer_bBot · 2023-08-16
> >
> > Many thanks for the response.
> >
> > In [W1] I meant Logistic Regression. I understand that the focus is only on binary classification, but still SOTA results in binary classification tabular datasets are achieved by Decision Tree methods (GBDT).
> >
> > I remain my positive review and consider increasing it.

---

> > > ### Author Response · Authors · 2023-08-19
> > > **Thanks for your further clarification**
> > >
> > > Thanks for your further clarification. Our method is designed for neural networks and can not be used for decision tree methods (e.g., XGBoost, GBDT). The key reason is that our algorithm (See Appendix G) involves gradient computation and weight perturbation (See Eqs. (23) and (25)) to accelerate the bi-level optimization problem-solving. We summarize the details as follows:
> > >
> > > * **Nature of Parameters**: In a neural network, the parameters are continuous values (weights, biases) that are updated using gradients to minimize the loss. In GBDT or XGBoost, the **"parameters" are the structure of the trees themselves**, including the split points and leaf values. These are not continuous values that can be updated with gradient descent in the traditional sense.
> > > * **Training Mechanism**: While GBDT and XGBoost use the concept of gradients (specifically, gradient boosting works by fitting new trees to the negative gradient of the loss), this is not the same as computing gradients with respect to weights in a neural network and updating them. Instead, **trees are added to the model** to correct the errors (**residuals**) of the current ensemble.
> > > * **Non-Differentiability**: Decision trees involve making decisions based on **hard thresholds**, which are inherently non-differentiable operations. This makes them unsuitable for traditional gradient-based optimization.
> > >
> > > Therefore, we leave the extension of this work for decision tree-based models in future work. We will provide a discussion on the decision tree in the revision.

---

### Official Review · Reviewer_Viye · 2023-07-07

**Soundness:** 3 good
**Presentation:** 3 good
**Contribution:** 2 fair
**Rating:** 6
**Confidence:** 2

**Summary:**

theoretical proof: prove that distribution shifting can be written as data perturbation, and the model trained with the perturbed data can be written as the perturbed model trained with original source data.
methodology: motivated by the connection between distribution shifting and data weight perturbation, the author proposed RFR(robust fairness regularization), which is an upper bound of the difference of demographic parity between source and target.

**Strengths:**

well written; easy to follow.
well-structured and coherent framework that facilitates easy comprehension.

**Weaknesses:**

- I think that the objective function which the author suggested did not reflect the true effect of $L_{RFR}$. Please add the experiments without $L_{DP}$
- It would be better if the notation $\delta_X, \delta_Y$ were changed. The author writes $\delta_X, \delta_Y$ to denote the perturbation for feature and label respectively, however, in equation (3), it seems like they depend on each random variable X and Y. It can make confusion for readers.
- some typos: for example, line 495(appendix), $E_{\delta_Y}[\delta_X]$ → $E_{\delta_X}[\delta_X]$, line 152 $\|\cdot\|_2$ →$\| \cdot\|_p$

**Questions:**

- I cannot understand the proof of theorem 2.3, lines 499-501; is that sufficient? Please give more details.
- I think it would be better if the graph of EO was also included in appendix.


**Limitations:**

refer weakness and questions.

---

> ### Author Rebuttal · Authors · 2023-08-08
>
> We thank the reviewer for the constructive comments.
>
> **[W1] I think that the objective function which the author suggested did not reflect the true effect of $L_{RFR}$. Please add the experiments without $L_{DP}$.**
>
> **Ans:** We thank the reviewer for this constructive comment. As required by the reviewer, we add ablation studies to reflect the true effort of $L_{RFR}$ in Figure 2 of the one-page rebuttal.
>
> **[W2] It would be better if the notation $\delta_X$, $\delta_Y$ were changed. The author writes $\delta_X$, $\delta_Y$ to denote the perturbation for feature and label respectively, however, in equation (3), it seems like they depend on each random variable X and Y. It can make confusion for readers.**
>
> **Ans:** We thank the reviewer for this thoughtful comment. As required by the reviewer, we will revise the notation $\delta_X$, $\delta_Y$ into $\delta_X(X)$, $\delta_Y(Y)$ for the first appearance and explicitly mentioned using $\delta_X$, $\delta_Y$ for simplicity for the remaining part of the paper.
>
> **[W3] Some typos: for example, line 495(appendix),
>  $E_{\delta_Y}[\delta_X] \rightarrow E_{\delta_{X}}[\delta_{X}]$, line 152 $|\cdot|_2 \rightarrow |\cdot|_p$.**
>
>  **Ans:** We will revise these typos in the revised version.
>
> **[Q1] I cannot understand the proof of theorem 2.3, lines 499-501; is that sufficient? Please give more details.**
>
> **Ans:** Eqs. (18) and (20) provide the loss value analysis for data and model weight perturbation. When chasing Eq. (18) equals Eq. (20), the model weight perturbation is treated as multivariate but with only one equation. It is easy to see that there exists a model weight perturbation solution to guarantee that Eq. (18) equals Eq. (20).
>
> **[Q2] I think it would be better if the graph of EO was also included in appendix.**
>
> **Ans:** We have added the tradeoff results between EO and accuracy in Figure 4 of the one-page rebuttal. We will add the results in Appendix for the revised version.

---

> > ### Comment · Reviewer_Viye · 2023-08-17
> > **elaborate quotation for author**
> >
> > for Q1) I updated my quotation before rebuttal period, but unfortunately it seems not visible for authors. Here is my updated quation:
> > It seems to me that additional conditions, such as non-zero, might be necessary for this proof to be sufficient. Could you explain the reasons behind the sufficiency of this proof?
> >
> > If these questions are addressed satisfactorily, I would be more than willing to increase my score.

---

> > > ### Author Response · Authors · 2023-08-17
> > > **Thanks for the valuable feedback**
> > >
> > > Thanks for the valuable feedback and the further clarification on Q1.  When chasing Eq. (18) equals Eq. (20), the model weight perturbation is treated as multivariate but with only one equation. In other words, considering linear equation $\mathbf{A}\mathbf{x}=b$, where $\mathbf{A}\in\mathbb{R}^{1\times n}$ and $b\in\mathbb{R}^{1\times 1}$ are both constant,  $\mathbf{x}\in\mathbb{R}^{n\times 1}$ is variables, the goal is to find whether the solution $\mathbf{x}$ exists for linear equation $\mathbf{A}\mathbf{x}=b$. Note that we consider distribution shift problem, i.e., $b\neq 0$, the solution for linear equation $\mathbf{A}\mathbf{x}=\mathbf{b}$ exists when $rank([A|b])=1=rank(A)$, i.e.,  $|A|=\bigg|E_{(X, Y)\sim \mathcal{P_S} }   [\frac{\partial l}{\partial f}\frac{\partial f}{\partial \theta}]\big|_{\theta=\hat{\theta}}\bigg|>0$. Note that such a **non-zero gradient condition** is easily satisfied for models that are not well-trained. We will add such a condition in the revision. Thanks again for the insightful suggestions.

---

> > > > ### Comment · Reviewer_Viye · 2023-08-17
> > > > **Updated my score**
> > > >
> > > > Your paper is well-structured, and some of my concerns have been addressed. I update my score.

---

### Official Review · Reviewer_iSrC · 2023-07-07

**Soundness:** 2 fair
**Presentation:** 4 excellent
**Contribution:** 2 fair
**Rating:** 6
**Confidence:** 3

**Summary:**

In this work, the authors seek to learn neural networks that are fair not just w.r.t. the observed distribution of individuals, but also under shifts to this distribution. In order to achieve this, they propose robust fairness regularization (RFR) which uses adversarial weight-space perturbations at train time to create more robustly fair models (w.r.t. distribution shifts). RFR being a reliable regularizer for distribution shifts relies on a correspondence between distribution and point-wise data perturbations as well as a correspondence between point-wise data perturbations and model weight perturbations. Finally, the paper proposes that for each group, the difference between the predictive performances on the source and target distribution should be the same. By enforcing this over all groups, the authors have the RFR loss that they propose.

**Strengths:**

The paper focuses on a very important topic for trustworthy ML.

The paper is well-written and the notation is clear and easy to follow.

The experimental results that are reported are strong and at the core evaluate the method well, though I have some suggestions below in the weaknesses for improvements here.

**Weaknesses:**

Major weakness: The major weakness that I see with this paper is the underlying theoretical motivation. In particular Theorem 2.3 appears to have a critical flaw that is never addressed in the paper (as far as I can tell). I elaborate further at the bottom of this section, but the key point is this, though there exists weight perturbation that give us the same loss as any arbitrary input perturbations, there is no given relationship between the magnitude of the perturbations in either domain. Therefore, even given that you know the Wasserstein distance is bounded by some value $\epsilon$ there is no way to select the value of $\rho$ before you have observed the target distribution, rendering the method difficult to use in practice. Moreover, I find it troubling that this is (1) not addressed in the paper, and (2) the authors do not explain how they select $\rho$ either theoretically or experimentally. It is obvious to me that given access to the target distribution, one can optimize $\rho$ to give good results, but this is the same as training on your test distribution and claiming good results. Without a clear and theoretically grounded argument for selecting $\rho$ and an open source implementation showing that this works well across datasets, I do not think I can vote to accept this paper.


Another weakness I see, is that the authors do not compare their weight perturbation method to the corresponding input perturbation method (as far as I can see). The ADV method used in the experiments refers to methods like LAFTR which are adversarial, but not in the same spirit as crafting worst-case perturbations.


Elaborated major weakness: The  first claim (Corollary 2.2)  is that there exists a perturbation map that makes the classification loss between source, $P_S$, and target $P_T$ the same. This statement seems clear and intuitive; however, we do not, a priori know how the magnitude by which we will have to perturb each data point. Let us assume, for sake of argument, that we know a priori that the Wasserstein distance between $P_S$ and $P_T$ is bounded by a real value $\epsilon > 0$. Then for an empirical distribution consisting of $n$ atoms, we know that we won’t perturb any given point by more than $n\epsilon$, so this gives us some practical guidance on how to set the data perturbation value, assuming we know a bound on the Wasserstein distance. The second claim made by the paper, Theorem 2.3, is that there exists a weight perturbation such that the model has the same loss on the source distribution (after its weights have been perturbed) as it does on the target distribution. However, though we may have a bound on how large the input perturbation may be, there is absolutely no correspondence between the magnitude of input perturbations, $n\epsilon$ and the magnitude we should set for the weight space perturbations $\rho$.

**Questions:**

Please address my key point on how one picks $\rho$ in a principled way.

**Limitations:**

See weaknesses.

---

> ### Author Rebuttal · Authors · 2023-08-08
>
> We thank the reviewer for the constructive comments.
>
> **[Q1] The major weakness that I see with this paper is the underlying theoretical motivation. Theorem 2.3 appears to have a critical flaw. (1) There is no given relationship between the magnitude of the perturbations in either domain and thus there is no way to select the value before you have observed the target distribution. (2) The authors do not explain how they select either theoretically or experimentally.**
>
> **Ans:** We thank the reviewer for this constructive comment.
>
> (1) We agree with the reviewer that there is no given relation between the magnitude of the perturbations $\rho$ in either domain. However, we would like to respectfully clarify that it is not necessary to consider such a relation since there is no accessible information on target distribution. As shown in Table 1 of our paper, a heuristic $\rho=0.0005$ contributes to a better accuracy-fairness tradeoff than baseline methods.
>
> (2) We would like to respectfully clarify that it is unnecessary to theoretically or experimentally optimize $\rho$. According to the result of the hyperparameter study on $\rho$ given $\lambda=0.1$ in Figure~4 of our paper, the accuracy and DP are stable for $1\times 10^{-5} \leq \rho\leq1\times 10^{-3}$, which indicates the performance is rarely affected by the selection of $\rho$. A heuristic $\rho=0.0005$ in our experiment is sufficient for mitigating the unfairness under distribution shift.
>
> As Reviewer #3kzb noted, *more discussion of the hyper-parameter $\rho$ is not necessary for this work*, given the extensive experimentation in this paper. This indicates our analysis and experiment sufficiently demonstrate the effectiveness of RFR.
>
>
> **[Q2] The authors do not compare their weight perturbation method to the corresponding input perturbation method (as far as I can see). The ADV method used in the experiments refers to methods like LAFTR which are adversarial, but not in the same spirit as crafting worst-case perturbations.**
>
> **Ans:** Thanks for the insightful comments. We agree that ADV is an adversarial method like LAFTR and different from the input perturbation method. However, the input perturbation method is complicated since the input perturbation is dependent on input samples. Meanwhile, the proposed model weight perturbation is simple and the weight perturbation for all samples is the same. Considering the worst-case scenario, the input adversarial method is similar to adversarial attack while the goal of the attack is different. The method of perturbation/attack is highly related to feature type (numerical, categorial, or mixed). We leave the investigation of the input perturbation method for future work.

---

> > ### Comment · Reviewer_iSrC · 2023-08-14
> > **Thank you for your response**
> >
> > I would like to thank the authors for their response. I found their answers somewhat satisfactory, but felt that my question went largely unanswered. I understand that a hyper-parameter search can be done to optimize $\rho$ (which is how the authors came up with $\rho = 0.0005$, but this hyper-parameter search requires that one has access to the target distribution of interest (in order to validate the best range of $\rho$ values). Also $\rho$ is the critical parameter in Equation (8), (9), and (10) i.e., the core contribution of the work so selection of this parameter should not be swept under the rug.
> >
> > If the authors could clearly answer this I would be more satisfied: given $n$ samples from the source distribution for training and 0 samples from the target distribution, how do I empirically select $\rho$? Should I always set $\rho = 0.0005$ no matter what network architecture or dataset I am using? I find it hard to believe that even a simple fully connected network that has width 8 and a fully connected network that has width 4096 will have the same optimal $\rho$ value given the disparity in the weight magnitude distributions.
> >
> > Also have I misunderstood Figure 4? Where is the abalation on $\rho$? I only see variations on $\lambda$ (the x-axis on both plots) and no variation of $\rho$.
> >
> > Nonetheless, the problem studied is an interesting and important one and the paper reports encouraging experimental results which may be built upon by the community in future. I would like to increase my score, but need further clarification from the authors on these points.

---

> > > ### Author Response · Authors · 2023-08-15
> > > **Thank you for your response**
> > >
> > > **[Q1] If the authors could clearly answer this I would be more satisfied: given $n$ samples from the source distribution for training and $0$ samples from the target distribution, how do I empirically select $\rho$
> > > ? Should I always set $\rho=0.0005$ no matter what network architecture or dataset I am using? I find it hard to believe that even a simple fully connected network that has width $8$ and a fully connected network that has width $4096$ will have the same optimal value given the disparity in the weight magnitude distributions.**
> > >
> > > Thanks for your constructive comments. We agree that it is intractable to select the optimal $\rho$ without accessing target data distribution. In our experiments, we **heuristically select** $\rho=0.0005$ and it works well. We also agree that the optimal $\rho$ is dependent on network architecture and dataset, which indicates hyperparameter tuning may further boost performance.
> > >
> > > For the setting without accessing target distribution, the only feasible way, to the best of our knowledge, is to use training distribution for validation, although it may not be an optimal choice. However, if we can access more information (e.g., few-shot target samples or input features in the target dataset), the hyperparameter tuning can be done better. For example, if the input features in **(unlabeled) target dataset** are available, it is tractable to predict out-of-distribution error and then use it for hyperparameter selection [1,2]. Note that the input feature in the target dataset should be available in the inference stage.
> > >
> > > [1] Yu, Yaodong, et al. "Predicting out-of-distribution error with the projection norm." International Conference on Machine Learning. PMLR, 2022.
> > >
> > > [2] Lu, Yuzhe, et al. “Predicting Out-of-Distribution Error with Confidence Optimal Transport.” arXiv preprint arXiv:2302.05018 (2023).
> > >
> > > **[Q2] Also have I misunderstood Figure 4? Where is the ablation on $\rho$? I only see variations on $\lambda$ (the x-axis on both plots) and no variation of $\rho$.**
> > >
> > > Sorry for the typo in the previous response. Please see **Figure 1** for the hyperparameter study on $\rho$ in the **one-page rebuttal** at general comments https://openreview.net/forum?id=DVjyq5eCAD&noteId=wRMAbkBVic.

---

> > > > ### Comment · Reviewer_iSrC · 2023-08-15
> > > > **Thank you for clarification!**
> > > >
> > > > I would like to thank the authors for their clear and quick response to my comment. Figures 1 and 2 of the supplied results are very helpful as were the supplied papers. I have subsequently increased my score. As a final note, when adding Figures 1 and 2 to the paper, please add the $\rho = 0$ case to complete the figure. Thanks again for clearing up my doubts!

---

> > > > > ### Author Response · Authors · 2023-08-15
> > > > > **Thanks for increasing the score**
> > > > >
> > > > > Thank you for your valuable feedback and for significantly increasing the score to 6! We will incorporate the case of $\rho=0$ to complete the figure, and the discussion during rebuttal in our revision.

---

### Official Review · Reviewer_646Z · 2023-07-27

**Soundness:** 3 good
**Presentation:** 3 good
**Contribution:** 3 good
**Rating:** 6
**Confidence:** 5

**Summary:**

The paper studies the problem of training a fair ML model in the presence of distribution shift. The proposed method aims to bound the biases on the target distribution and use it as a regularizer.


**Strengths:**

S1 The problem of distribution shift is quite fundamental.
S2 The idea of bounding the biases and using it as a regularization term is underexplored.


**Weaknesses:**

W1 The approach of using the worst-case bound as a regularizer has been explored before for selection bias [1]. Please discuss the differences with this approach.

W2 The experiments do not explain the observations in detail. Specifically, the variance is quite high for spatial shift. Are the results staistically significant? Why are the baselines doing worse?

It would be worth adding an experiment where source and target are exact same. I would expect the baselines to perform much better in this case. More generally, only when the source and target are very different, I expect RFR to perform better than the rest. The current experiments do not seem to clarify any such aspect.

W3 Please show how loose are the estimated bounds. Are the bounds tight?

W4 If possible, consider adding proofs or sketch of theorems.
[1] https://arxiv.org/pdf/2212.10839.pdf





**Questions:**

Why does RFR not have the best fairness for spatial distribution shift? Is it because lambda is small?


**Limitations:**

The paper does not discuss the limitations of the approach. I think the loose bound of fairness estimates can adversely affect model accuracy. Authors should discuss the practical implications of the presented approach.

---

> ### Author Rebuttal · Authors · 2023-08-08
>
> We thank the reviewer for the constructive comments.
>
> **[W1] The approach of using the worst-case bound as a regularizer has been explored before for selection bias [1]. Please discuss the differences with this approach.**
>
> **Ans:** We will add the discussion on the difference with [1]. The difference is two-fold: (1) The problem setting. [1] considers fairness problem with selection bias, where the selection bias is described with available auxiliary information. Our paper mainly focuses on fairness problem under distribution shift without any information on target distribution, which can be but is not necessarily caused by selection bias. (2) Methodology. [1] mainly focus on Consistent Range Approximation (CRA) of a fairness query using probability information in data collection. Our paper mainly focuses on DP metric difference in source and target distribution, and then further derives a model perturbation approach to achieve fairness under distribution.
>
> [1] https://arxiv.org/pdf/2212.10839.pdf
>
>
> **[W2] The experiments do not explain the observations in detail. Specifically, the variance is quite high for spatial shift. Are the results staistically significant? Why are the baselines doing worse? It would be worth adding an experiment where source and target are exact same. I would expect the baselines to perform much better in this case. More generally, only when the source and target are very different, I expect RFR to perform better than the rest. The current experiments do not seem to clarify any such aspect.**
>
> **Ans:** (1) The variance for spatial shift on ACS-I is higher than that of temporal shift on all methods, which indicates neural networks easily converge to different local minima for spatial distribution shift. We run each experiment five times and compare the average results. The proposed methods can achieve better fairness results for most cases and the tradeoff results clearly demonstrate the effectiveness. (2) For the setting without distribution shift, we add the experimental results in Figure. 3 and Table. 1. It is seen that adding regularization plays a strong baseline and achieves comparable performance with our proposed method. We will add the experimental results and more discussion in the revised version.
>
> **[W3] Please show how loose are the estimated bounds. Are the bounds tight?**
>
> **Ans:** The bound in Eq. (5) is tight when both condition (a) $DP_{\mathcal{S}}\leq DP_{\mathcal{T}}$ and (b) maximum and minimum of value set $\{ \mathbb{E}_{\mathcal{S}_0}[f_{\theta}(\mathbf{x})], \mathbb{E}_{\mathcal{S}_1}[f_{\theta}(\mathbf{x})], \mathbb{E}_{\mathcal{T}_0}[f_{\theta}(\mathbf{x})], \mathbb{E}_{\mathcal{T}_1}[f_{\theta}(\mathbf{x})] \}$ both are from source or target distribution. Even though conditions (a) and (b) may not hold for neural network model, we would like to that our goal is not to obtain a tight bound for demographic parity on target distribution. Instead, we aim to find **sufficient conditions** to guarantee low demographic parity and such low demographic parity can be achieved in model training without any target distribution information. The proposed upper bound Eq. (5) actually reveals sufficient conditions, which can be achieved by our proposed RFR algorithm.
>
> **[W4] If possible, consider adding proofs or sketch of theorems.**
>
> **Ans:** We will add the sketch of theorems in the revised version. Due to the main text page limitation, we will leave proofs in Appendices.
>
> **[Q1] Why does RFR not have the best fairness for spatial distribution shift? Is it because lambda is small?**
>
> **Ans:** As shown in the hyperparameter study (Figure. 4), hyperparameter $\lambda$ is critical to achieving a "good" trade-off between accuracy and demographic parity. When selecting a larger hyperparameter $\lambda$, RFR can achieve better fairness but worse accuracy. In this paper, we consider the worst case to tackle fairness problem under distribution shift without access to target distribution. Therefore, when the (unknown) target distribution closes (deviates) to the worst case, the fairness performance could be (not be) the best compared with several baselines. For spatial distribution shift, our proposed method can still achieve comparable or even better than ADV in terms of tradeoff performance, as shown in Figure 3.
>
>
> **[Q2]  I think the loose bound of fairness estimates can adversely affect model accuracy. The authors should discuss the practical implications of the presented approach.**
>
> **Ans:** The bound of fairness estimation in Eq. (5) is unrelated to accuracy performance. For the algorithm implementation, the accuracy and fairness tradeoff can be controlled by hyperparameter $\lambda$ as shown in Eq. (13). Please see Appendix G for practical implementation.

---

### Author Rebuttal · Authors · 2023-08-08

We thank all reviewers for the constructive comments and the positive evaluation for most reviewers. We provide the summary for the key revision.
* Add experiments on hyperparameter study on perturbation $\rho$.
* Add comparison results (Table. 1 and Figure 1 in one-page rebuttal) for the setting without distribution shift.
* Add more details on hyperparameter $\rho$ selection. We choose $\rho=0.0005$ in the experiments and it already works well. We also add discussion on several follow-up works to further improve our performance via better hyperparameter $\rho$ selection via auxiliary information or AutoML with target performance predictor.
* Add discussion on the difference from several existing works [1,2,3].
* Provide more justifications for experimental observation.
* More clarification on the bound Eq. (5).
* add more details on the proof of theorem 2.3, lines 499-501.
* Revise several notation typos.

[1] Zhu, Jiongli, et al. "Consistent Range Approximation for Fair Predictive Modeling" arXiv preprint arXiv:2212.10839.

[2] Wiles, Olivia, et al. “A Fine-Grained Analysis on Distribution Shift.” International Conference on Learning Representations. 2022.

[3] Lu, Yuzhe, et al. "Predicting Out-of-Distribution Error with Confidence Optimal Transport." arXiv preprint arXiv:2302.05018 (2023).

---

### Decision · Program_Chairs · 2023-09-21

**Decision:**

Accept (poster)

**Comment:**

This paper has an overall positive opinions after a couple of score revisions after rebuttal.

Authors propose a novel method to address fairness issues under covariate shift when target samples are not available. The problem is important. Authors did a number of additional experiments that include ablations, hyper parameter sensitivity and comparisons with baselines during the rebuttal that seems to have addressed reviewer concerns adequately.

I encourage the authors to add these new experimental results to the camera ready.